



# Methane releases across the Laptev Sea signaled by time-integrated biomarkers of aerobic methane oxidation

Albin Eriksson[1,2], Birgit Wild[1,2], Wei-Li Hong[2,3,4], Henry Holmstrand[1], Francisco J.A Nascimento[4,5], Stefano Bonaglia[6], Denis Kosmach[7], Igor Semiletov[7,8], Natalia Shakhova[7,8,9], and Örjan Gustafsson[1,2]

[1] Department of Environmental Science (ACES), Stockholm University, Stockholm, Sweden
[2] Bolin Centre for Climate Research, Stockholm, Sweden
[3] Department of Geological Sciences (IGV), Stockholm University, Stockholm, Sweden
[4] Baltic Sea Centre, Stockholm University, Stockholm, Sweden
[5] Department of Ecology, Environment and Plant Sciences, Stockholm University, Stockholm, Sweden
[6] Department of Marine Sciences, University of Gothenburg, Gothenburg, Sweden
[7] Laboratory for Arctic Research, V.I. Ilichov Pacific Oceanological Institute (POI), Far Eastern Branch of the Russian Academy of Sciences, Vladivostok, Russia
[8] Laboratory for Integrated Research of the Arctic System "land-shelf", National Tomsk State Research University (TSU), Tomsk, Russia
[9] Sadovsky Institute of Geosphere Dynamics (IGD) of the Russian Academy of Sciences, Moscow, Russia

*Correspondence to*: Albin Eriksson (albin.eriksson@aces.su.se) and Örjan Gustafsson (orjan.gustafsson@aces.su.se)

**Abstract.** Elevated methane concentrations in seawater have been reported over extensive areas of the East Siberian Arctic Seas, overlying thawing subsea permafrost. However, observed methane concentrations of the ephemeral seawater are highly variable across both space and time, compromised by both the timing of rare measurements and storm-driven exchanges to the atmosphere. Here, we applied time-integrated signals of the $\delta^{13}$C-composition of specific $C_{30}$ hopanoids (diploptene, hop-17(21)-ene, neohop-13(18)-ene and diplopterol) in surface sediments to trace aerobic methane oxidation and thereby provide a proxy for methane release. Interpretations of hopanoids and possible sources were further assessed by 16S-rRNA analyses in the surface sediments. The consistently low $\delta^{13}$C-$C_{30}$ hopenes signals, ranging between -57.5 to -37.1 ‰ (n=23) across the Laptev Sea shelf indicated aerobic methane oxidation. This suggests ubiquitous methane release with the most pronounced intensities in the outer shelf region, broadly consistent with the observed methane concentrations. Notably, depleted $\delta^{13}$C-$C_{30}$ hopenes were also found in the mid-shelf region of the Laptev Sea, earlier thought to be an area of comparatively low methane emissions. High methane concentrations were also observed in the vicinity of the Lena River delta, yet the isotopically heavier $\delta^{13}$C-$C_{30}$ hopenes may here reflect a combination of lower aerobic methane oxidation, a greater relative abundance of type II methanotrophs (lower isotope fractionation during hopanoid production) and isotope dilution from non-methanotrophic sources. While this complicates the biomarker interpretation in the unique setting near the Lena River delta, the $\delta^{13}$C-$C_{30}$ hopenes were still much lower than $\delta^{13}$C-OC, indicating aerobic methane oxidation and a clear methane release signal also in this regime. Taken together, the results unravel the wider cross-shelf patterns of methane releases in the Laptev Sea through probing of methane fossilised in membrane lipids of aerobic methanotrophs with the molecular-isotopic pattern being preserved in the sedimentary archive.



## 1 Introduction

Large uncertainties remain regarding methane ($CH_4$) emissions from both anthropogenic and natural systems (Saunois et al., 2025 Schaeffer et al., 2025). Terrestrial permafrost stores massive amounts of carbon, but climate change and Arctic
amplification have enhanced its thawing, causing net $CH_4$ emissions to the atmosphere (Ramage et al., 2024; Hugelius et al., 2024). Subsea permafrost, formed from the inundation of the northern tundra during the last deglaciation, is often overlooked, in part due to its inaccessibility, limiting the number of observations. However, observations suggest highly elevated $CH_4$ concentrations in the shallow seawater column overlying subsea permafrost (e.g., Semiletov, 1999; Shakhova et al., 2010; Steinbach et al., 2021) that it is currently thawing about ten times faster than nearby terrestrial permafrost in Northeastern
Siberia (Shakhova et al., 2017; Wild et al., 2022).

The East Siberian Arctic Shelf (ESAS), the world's largest continental shelf sea, is very shallow (on average, ca. 50 m) and estimated to host the majority of the world's subsea permafrost (Romanovskii et al., 2001; Romanovskii et al., 2005; Overduin et al., 2019). The concentration of $CH_4$ in the surface waters of the ESAS is, particularly in the Laptev Sea, much higher than
what would be expected from $CH_4$ levels in the atmospheric boundary layer, suggesting that the thawing subsea permafrost system may serve as a conduit for $CH_4$ release to the atmosphere (Shakhova et al., 2010, 2014, 2019). A key challenge to constrain the geospatial pattern of $CH_4$ releases is the short-term variability of water column $CH_4$ concentrations, together with the sparseness of observations over time in this remote region. While subsea permafrost thaw and related $CH_4$ emissions are governed by environmental changes on decadal time-scales (Shakhova et al., 2017, 2019), the sparse and infrequent
measurements of water column $CH_4$ concentrations might capture neither temporal nor geospatial trends of $CH_4$ emissions from these systems. Water sampling has revealed the general pattern of elevated $CH_4$ concentrations and discovered a few major hotspots, but this pattern is strongly affected by short-term variability governed by storms and the shallow water column of the Laptev Sea (e.g., Shakhova et al., 2014). Furthermore, a recent modeling study concluded that current atmospheric observation networks are not able to detect abrupt $CH_4$ releases from this region (Wittig et al., 2024). To better grasp the $CH_4$
dynamics over a longer timespan than what is captured in snapshot seawater samples, there is a need for an integrated metric to observe patterns of long term $CH_4$ release from the subsea permafrost system over both space and time scales.

Specific biomarkers in surface sediments offer an opportunity to study large-scale variabilities in $CH_4$ and its dynamics integrated over time scales of years to decades. Bacteria in oxygenated sediments and in the water column can utilize $CH_4$ as
an energy and carbon source through aerobic $CH_4$ oxidation (AeOM), converting it to carbon dioxide ($CO_2$; Hanson & Hanson, 1996), or assimilating it into biomass (e.g., lipid synthesis). Hopanoid lipids such as diploptene and diplopterol are important constituents of the cell membranes of Methane Oxidizing Bacteria (MOB) carrying out these processes (Summons et al., 1994), but can also derive from bacteriohopanepolyols (BHPs) upon diagenesis (Mackenzie et al., 1981). Hopanoids and BHPs are not only produced by MOB, but can stem from a variety of aerobic and anaerobic bacteria utilizing $CH_4$ and/or





inorganic/organic carbon (Rohmer et al., 1980; Summons et al., 1999; Sinninghe Damsté et al., 2004; Belin et al., 2018; Grinko et al., 2020). Thus, hopanoid analyses are frequently combined with compound-specific isotope analysis of stable carbon isotopes ($\delta^{13}C$) to relate these biomarkers to $CH_4$ cycling by utilizing the low, i.e., isotopically depleted, $\delta^{13}C$-$CH_4$ (as low as -90 ‰; Milkov & Etiope, 2018) to differentiate $CH_4$-derived hopanoids from other sources (e.g., Hinrichs, 2001; Hinrichs et al., 2003; Birgel and Peckmann, 2008; Davies et al., 2016; Inglis et al., 2019; Sun et al., 2022; Blumenberg et al., 2024; Yan

et al., 2025). Generally, a larger presence of $CH_4$ has been linked to decreasing $\delta^{13}C$ values of hopanoids (e.g., van Winden et al., 2020). Therefore, $\delta^{13}C$-hopanoids may be used as indicators of the intensity of a time-integrated $CH_4$ release signal. In particular, hopanoid chain lengths $\leq C_{30}$ are generally more depleted in $^{13}C$ compared to $C_{31}$-hopanoids and therefore closely associated with MOB production of hopanoids (Inglis et al., 2019). Contemporary system calibrations of $\leq C_{30}$ hopanoids exist in peatlands (Inglis et al., 2019) and lacustrine systems (Davies et al., 2016). However, the lack of a large-scale comparison of

$CH_4$ and hopanoids in marine systems leaves uncertainties for interpreting $\delta^{13}C$-hopanoids in geological records. In summary, $\delta^{13}C$-hopanoids in oxygenated surface sediments can be an informative tool to constrain a time-integrated $CH_4$ release signal, complementing observations of the highly variable $CH_4$ concentrations in the water column.

Observations across the Laptev Sea suggest the presence of AeOM, but its efficiency is questionable and spatially variable.

The Laptev Sea water and surface sediments are well oxygenated (Stepanova et al., 2017; Brüchert et al., 2018; Maciute et al., 2025), indicating the general possibility for AeOM. Furthermore, microbial studies have confirmed the presence of MOB in Laptev Sea sediments and water column (Samylina et al., 2021; Savvichev et al., 2023). Recent machine-learning estimations deemed the global AeOM to be generally efficient in coastal waters (Mao et al., 2022), but AeOM observations in the vicinity of the Lena River delta display a rather weak methanotrophic sink, with a $CH_4$ turnover time of 193 days in mixed sea/river

water (Bussmann et al., 2017). In contrast, incubation experiments in the outer Laptev Sea cold seep region suggest more efficient AeOM in those surface sediments (Tikhonova et al., 2021), but with very low AeOM rates in the water column (300-1000 days turnover time, Shakhova et al., 2015). Taken together, AeOM is active in the Laptev Sea, but observations are diverging on the scale and efficiency of AeOM across the Laptev Sea. Such discrepancies highlight the importance of constraining the geospatial distribution of AeOM to unravel areas of more intense $CH_4$ cycling.


In this study, we explore AeOM biomarkers as time-integrated proxies of $CH_4$ release from Arctic Ocean sediments, focusing on the Laptev Sea. We hypothesize that known $CH_4$ ebullition hotspots in the Outer Laptev Sea and Inner Laptev Sea (OLS and ILS), where >1000 nM dissolved $CH_4$ concentrations have been observed (Shakhova et al., 2010, 2014; Steinbach et al., 2021), have high concentrations and low $\delta^{13}C$ values of AeOM-tracing $C_{30}$ hopanoids (diploptene, hop-17(21)-ene, neohop-

13(18)-ene and diplopterol). In contrast, we hypothesize that the mid-shelf region without any discovered $CH_4$ hotspots, yet with dissolved methane in the range 10-60 nM (Fig. 2) display lower concentrations of higher $\delta^{13}C$-hopanoids. Observations of AeOM biomarkers are also compared to water $CH_4$ concentrations and 16S-rRNA data on abundance of MOB- and non-methanotrophic hopanoid-producing bacteria. We finally use this dataset to explore the geospatial distribution of $CH_4$ release



across the 600 km wide Laptev Sea shelf. Insights from this study also allows to establish some general considerations for interpreting hopanoids in geological records.

## 2 Methods

### 2.1 Study area and the ISSS-2020 expedition

During the last deglaciation, sea level rise caused the northern regions of East Siberian permafrost to be submerged, thus forming the present coverage of Laptev Sea subsea permafrost (Romanovskii et al., 2001; Romanovskii et al., 2005; Nicolsky & Shakhova, 2010; Shakhova et al., 2017; Lindgren et al., 2018; Overduin et al., 2019). The Laptev Sea water column is shallow (average water depth ~50 m), with well oxygenated top mm to cm of surface sediments (Brüchert et al., 2018; Maciute et al., 2025). Ebullition appears to be a key transport vector in this shallow region, transferring large amounts of $CH_4$ to the water column (Shakhova et al., 2010, 2014; Steinbach et al., 2021). The Inner (ILS; latitudes < 73.9°N & longitudes ~130°E) and Outer Laptev Sea hotspot regions (OLS; latitudes ≥ 76.1°N & longitudes 125-131°E) are established as areas of extensive $CH_4$ activity in the water column, with their shallowness causing frequent ventilation of the water column $CH_4$ to the overlying atmosphere (e.g., Shakhova et al., 2010, 2014; Thornton et al., 2016, 2020; Steinbach et al., 2021; Chernykh et al., 2023). Results from campaigns of subsea permafrost drilling have indicated rapid thawing of subsea permafrost in the ILS region (e.g., Shakhova et al., 2017). Thawing subsea permafrost is considered to facilitate $CH_4$ release from the very shallow ILS (Shakhova et al., 2014; Sapart et al., 2017; Wild et al., 2022). In the deeper, yet still shallow OLS (46-72 m), a thermogenic gas reservoir has been constrained as the main $CH_4$ source using triple isotope fingerprinting (Steinbach et al., 2021). The Laptev Sea is also strongly influenced by the Lena River plume (Shakhova & Semiletov, 2007; Shakhova et al., 2010). While the ILS receives substantial amounts of organic carbon from the Lena River (Vonk et al., 2012; Holmes et al., 2012; McClelland et al., 2016; Bröder et al., 2018; Martens et al., 2022), the riverine export of $CH_4$ is negligible in comparison to the regional $CH_4$ emissions (Shakhova & Semiletov, 2007; Shakhova et al., 2010). The mid-outer Laptev Sea transect (MLS; latitudes between 73.9-76.1°N & longitudes ~130°E) is, in contrast, an area of lower $CH_4$ concentrations than the ILS and OLS, yet the still elevated $CH_4$ levels over extensive MLS scales (Shakhova et al., 2010) call for investigations on potential $CH_4$ sources also for this region. Samples in the current study were collected from the Laptev Sea during the International Siberian Shelf Study in 2020 (ISSS-2020; Sept- Nov. 2020) aboard the research vessel *Akademik Keldysh* (Fig.1).





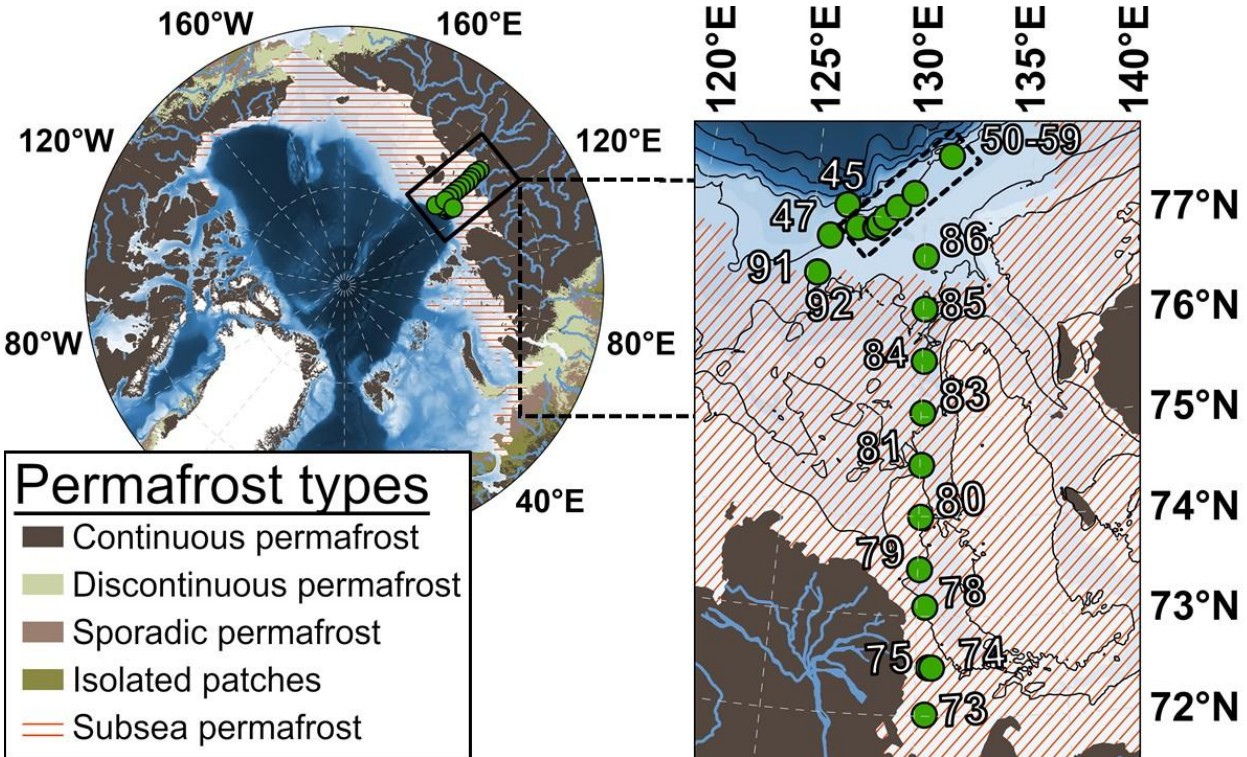

**Figure 1.** The Laptev Sea sampling locations and station numbers, including estimates of the current Arctic land and subsea permafrost covers. Permafrost datasets were generated from Obu et al. (2019) and Overduin et al. (2019). The bathymetry was plotted using IBCAO version 5.0 (Jakobsson et al., 2024).

## 2.2 Sediment and Water Sampling

### 2.2.1 Sediment sampling

Sediment samples were retrieved using an Oktopus Multicorer System that sampled eight cores per station with an undisturbed sediment-water interface. The penetration depth of the liners was a maximum of 30 cm. Overlying water was drained off the surface sediment in the cores and the sediment was subsequently sectioned in 1 cm intervals onboard using an Oktopus Core Extruder. All sediment subsamples for organic geochemical analyses were immediately frozen in plastic seal bags and the 1-2 cm slice in each core was used to extract lipid biomarkers for this study.

Sediment cores with pre-drilled holes on the liners were collected for 16S-rRNA analysis at n=36 stations (Maciute et al., 2025), where n=11 samples from the Laptev Sea shelf were used in this study. The overlying water was siphoned from the top of the sediment core and 3 ml of sediment was subsampled from three sampling depths in the core (0-1.5 cm, 1.5-3 cm, & 3-4.5 cm). Subsamples were vortexed with 10 ml RNA*later* (Sigma Aldrich) and stored for 2 h at 4°C before being frozen at -80°C. In this study, we only report the top 0-1.5 cm subsection for microbial analyses as the focus was on aerobic $CH_4$ oxidizers.





**2.2.2 Measurements of dissolved methane concentration in the water column**

Water samples were taken from a Rosette sampler with 24 x 10 L bottles, closed at selected depths. Subsamples were immediately taken through silicon tubes into 60 ml Luer-lock plastic syringes for onboard $CH_4$ concentration measurements. Briefly, a Restek reservoir bag was used for helium headspace equilibration with an IKA KS 260 benchtop shaker. Before

concentration measurements using Gas Chromatography-Flame Ionization Detection (GC-FID), the gases were transferred through stainless-steel Millipore filter holders packed with granular Drierite and quartz wool, sealed using PTFE tape. The $CH_4$ concentration measurements were made with 5 ml sample injections through one 1/8" Porapak Q packed pre-column and a 3 m 1/8" Heyesep D column before FID. In this study, the surface water is described as the topmost CTD/Rosette-based water samples (mean =3.5 m; n=21) and the sub-pycnocline (mean =37 m; n =71) as water samples below the largest density

gradient ($\sigma_\theta$) based on the CTD profile (Conductivity, Temperature and Depth). The sub-pycnocline concentrations are here reported as the median and interquartile ranges of all samples between the pycnocline and the bottom water (Fig. 2).

**2.3 Measurements of methane-tracing lipid biomarkers and their isotope composition**

Bulk organic carbon and stable carbon isotope ($\delta^{13}$C-OC) analyses of the 1-2 cm slice in each multicore were made according to standard operating procedures at Stockholm University (e.g., Martens et al., 2020). Briefly, freeze-dried sediment was

weighed in Ag capsules and acidified with 100 µl 1M HCl twice for the removal of inorganic carbon. After the first acidification, the sample was stored at 50°C for 3-4 hours, followed by acidification again and reaction at 50°C overnight. The acidified samples were wrapped in tin capsules and analysed with an elemental analyser coupled to an isotope ratio mass spectrometer (EA-irMS, Delta V Plus, Thermo Scientific, Germany).

**2.3.1 Lipid extraction**

Around 5 g of freeze-dried sediment was subsampled from one core per station. Before the extractions, internal standards were added to each sample (5α-androstane, tetracosane-$d_{50}$, triacontane-$d_{62}$, eicosanoic acid-$d_{39}$, 2-hexadecanol and stigmasterol-$d_5$) to calculate recoveries and correct for losses during the laboratory procedure. The lipid extraction and clean-up were a modification of standard operating procedures at Stockholm University (e.g., van Dongen et al., 2008; Vonk et al., 2012; Martens et al., 2020). Briefly, lipids were extracted using accelerated solvent extraction (ASE 350, Dionex), followed by

elemental sulphur and water removal from the extracts by addition of HCl-activated Cu and anhydrous $NaSO_4$. The total lipid extracts were saponified for 2 h at 80°C following the addition of ~3 ml KOH (6 % in MeOH) to each of the dried samples. Base-catalysed hydrolysis was applied as it has been shown to increase the amount of hopenes through dehydration of diplopterol (Sessions et al., 2013). The saponified neutral lipids (containing hopanoids, alkanes, sterols and alkenones) were collected with liquid-liquid extraction five times with the addition of 1 ml Milli-Q water and ~2 ml hexane: DCM (9:1, v/v).

After collecting the neutral fraction, the saponified sample was acidified to pH 1 with 6 M HCl, ~2 ml hexane: DCM (9:1, v/v) was added, the sample was mixed and the supernatant containing the fatty acid fraction was collected. The fatty acid liquid-



liquid extraction was repeated five times in total. In this study, we focused on the neutral fraction, containing hopanes/hopenes/diplopterol.

Following the saponification and separation of the neutral and fatty acid fractions, preparative chromatography was applied to the neutral fraction, with furnace-baked alkaline $Al_2O_3$ (~0.5 g) in a pre-combusted Pasteur pipette as the stationary phase. The neutral fraction was separated into non-polar hydrocarbons (hopanes/hopenes, F1), alkenones/diplopterol (F2) and polar hydrocarbons (F3) using 4 ml hexane: DCM (9:1, v/v), hexane: DCM (1:1) and DCM: MeOH (1:1, v/v), respectively. To enable clean chromatograms for $\delta^{13}C$ measurements, urea adduction was applied to purify the non-polar extracts (Pancost et

al., 2008; Inglis et al., 2019). The sample was separated into an adduct (containing *n*-alkanes) and a non-adduct fraction (containing cyclic and branched hydrocarbons, e.g., hopenes). Before instrument analysis, the diplopterol fraction was silylated for 1h at 80°C using BSTFA: pyridine (1:1).

### 2.3.2 Hopanoid identification and quantification

Hopanoids were quantified using gas chromatography-mass spectrometry (GC-MS 7820-A, Agilent Technologies USA) with
a DB5-MS ultra-inert column (30 m x 250 µm x 0.25 µm). The GC-MS was run in splitless mode with a starting temperature of 50°C for 2 min, then 10 °C/min to 210°C, where the temperature increased by 3°C/min to 262°C for 10 min. Thereafter, the temperature increased by 0.5°C/min to 270°C and lastly a 3°C/min ramp until 310°C for 5 min. A post-run at 305°C for 2 min was used to minimize sample carryover. Identification of hopanoids was done according to published mass spectra in Sessions et al. (2013), Sinninghe Damsté et al. (2014) and Elvert & Niemann, (2008). Hopenes were quantified using the $C_{30}$ hopane
external calibration standard (NIST 2266). However, during sample pre-treatment, dehydration of diplopterol and subsequent conversion to diploptene and hop-21-ene were observed within F2. This was likely catalysed by the basic $Al_2O_3$ column, as the problem resolved when switching to neutral $Al_2O_3$, which was used to separate the molecules before the final analysis. Diplopterol dehydration commonly occurs during sample workup, even during instrumental analysis (Kannenberg et al., 1995; Sessions et al., 2013). Thus, in this paper, we only present the $\delta^{13}C$ value of diplopterol when possible and not its
concentrations.

### 2.3.3 Compound-specific isotope analysis of hopanoids

Compound-specific isotope analysis (CSIA) of hopenes/diplopterol was carried out at Stockholm University using gas chromatography coupled to continuous flow isotope ratio mass spectrometry (GC-irMS; Trace-GC, GC-Isolink and Delta V Plus, Thermo Scientific, Germany) on a DB5-MS ultra-inert column (60 m x 0.25 mm x 0.25 µm). The oven temperature
program was the same as for GC-MS analyses. Briefly, triplicate 3 µl injections were made for each sample and quantified using external *n*-alkane standards (n-alkane mixture A7, Dr. Arndt Schindelmann, Indiana University). Before each batch of sample measurements, the precision and linearity of *n*-alkane standards were monitored to ensure precision ≤ 0.5 ‰ (n =6). The reproducibility of triplicate sample measurements was 0.1-2.3 ‰ (mean= 0.8 ‰). The higher standard deviations of some





samples were attributed to lower peak areas, causing less precise measurements. To minimize the precision effect from lower

peak areas, an acceptance threshold of $\geq$100 mV peak areas was set, at which *n*-alkane standard measurements yielded a precision of 0.9 ‰. The added trimethylsilyl (TMS; Si-(CH$_3$)$_3$) in $\delta^{13}$C measurements was corrected by measuring a silylated β-sitosterol standard using GC-irMS and the unsilylated β-sitosterol powder with EA-irMS. The difference in $\delta^{13}$C between the silylated and unsilylated standard was used as the $\delta^{13}$C of the added carbon to derivatized compounds, where f is the fraction of added carbon from the TMS group and $\delta^{13}$C$_{TMS}$ is the isotopic value of the TMS (Eq. 1).

$$\delta^{13}C_{Diplopterol} = \frac{(\delta^{13}C_{Silylated\ diplopterol} - f*\delta^{13}C_{TMS})}{(1-f)} \tag{1}$$

## 2.4 Microbiology

### 2.4.1 DNA extraction and sequencing

Sediment samples were thawed and prepared to facilitate 16S-rRNA analyses as described in Maciute et al. (2025). Amplification of the 16S rRNA gene V4 region was conducted with the primers 515 F (Parada et al., 2016) and 806 R (Apprill

et al., 2015). Library preparation was conducted using the NEBNext® Ultra™ II DNA Library Prep Kit with index adapters synthesized in-house by Novogene. The pooled library was sequenced on the Illumina NovaSeq 6000 SP platform with a 2x250 bp paired-end setup. The raw sequencing data have been uploaded to NCBI GenBank (https://www.ncbi.nlm.nih.gov/bioproject/ ) and can be accessed at BioProject: PRJNA1264051.

### 2.4.2 Bioinformatics

A total of 2,341,395 million sequencing reads, averaging 65 039 reads per sample (36 samples in total) were analysed according to the DADA2 (version 1.28) pipeline (Callahan et al., 2016) with default settings except for the filter commands: truncLen=c(0,0), maxEE=2, truncQ=2, trimLeft=c(22,21); error model: MAX_CONSIST = 30; merging of pair-ends: minOverlap=10; and chimera removal: allowOneOff=TRUE, minFoldParentOverAbundance=4. After trimming, filtering, merging pair-ends and chimera removal, a total of 89,148 amplicon sequence variants and 9.2 million sequences were retained,

averaging 43,485 reads per sample (minimum 26,424 and maximum 66,969 reads per sample). The sequences in the created ASV table were then assigned to taxonomy using the SILVA database version 132 (Quast et al., 2013) and analysed further as relative abundances (i.e., [counts/∑counts] × 100) in RStudio software. Only the samples collected and sequenced from the Laptev Sea shelf are discussed in this study (n = 11).

### 2.4.3 Relative abundance of methane-oxidizing bacteria and non-methanotrophic hopanoid producers

Microbial sequencing data were explored for both aerobic methanotrophs as well as other hopanoid-producing bacteria. Aerobic methanotrophs were identified within *Gammaproteobacteria* (Type I) and *Alphaproteobacteria* (Type II). Bacteria within phyla that commonly encode *shc* (squalene-hopane cyclases, i.e., hopanoid producers), such as *Cyanobacteria,* were



assumed to be hopanoid producers. Literature exploration of other organisms with documented production of hopanoids was made using LOTUS (Rutz et al., 2022) and was further used to assess "Other hopanoid producers". Particularly, the abundance
of certain *Acidobacteria*, *Actinobacteria* and non-methanotrophic *Proteobacteria* described in previous research of hopanoid producers (Sohlenkamp & Geiger, 2016; Sinninghe Damsté et al., 2017) was used to explore non-methanotrophic hopanoid producers. The information on all investigated hopanoid producers can be found in the supplementary information (Data availability).

**2.5 Statistical analyses**

Statistical differences were evaluated using a significance level α=0.05 in RStudio (Version 2024.12.1). One-way Welch's ANOVAs were applied to analyze differences in hopanoid concentrations and isotope signatures between the OLS, MLS and ILS due to the wide concentration ranges and the limited number of samples in the MLS. After Welch's ANOVAs, Games-Howell post-hoc tests were applied to verify where there were significant differences. Comparisons of $CH_4$ concentrations above and below the pycnocline were made using t-tests assuming unequal variances. All statistical results (F-values, degrees
of freedom, n-values, p-values and adjusted p-values) are reported in the Supplementary Tables S1-S5.

Sequencing data indicated three microbial types of hopanoid producers (Fig.S1&S2) and no evidence for anammox bacteria. Aerobic methanotrophs and heterotrophs were most common and therefore used as the two endmembers for hopanoid source apportionment. The hopanoid source apportionment was calculated through a Bayesian stable carbon isotope mixing model
using two endmembers (MOB & heterotrophic bacteria) applied with "simmr" in RStudio (Parnell et al., 2013). Due to a larger isotope fractionation in MOB-I hopanoid synthesis compared to MOB-II (Jahnke et al., 1999), the relative contribution of the different AeOM pathways was estimated using the relative abundance of MOB-I versus MOB-II within each sample (Eq.2). Since 16S-rRNA does not indicate the relative activity of these microbes, this approach is a semi-quantitative estimation of $CH_4$-related hopanoid production. The associated isotope endmembers and isotope fractionation during hopanoid biosynthesis
are displayed in Supplementary Tables S6 & S7 and further described in Eqs. 2-4, where $f_{MOB-I}$ and $f_{MOB-II}$ are the fractions of MOB based on the relative abundance of 16S-rRNA. The $\Delta\delta^{13}C_{MOB-I/II}$ and $\Delta\delta^{13}C_{DOC}$ describe the isotope fractionation from $CH_4$ to hopanoid and dissolved OC to hopanoid, respectively.

$$\delta^{13}C_{MOB} = f_{MOB-I} * (\delta^{13}C_{CH4} + \Delta\delta^{13}C_{MOB-I}) + f_{MOB-II} * (\delta^{13}C_{CH4} + \Delta\delta^{13}C_{MOB-II}) \qquad (2)$$

$$\delta^{13}C_{Heterotroph} = \delta^{13}C_{DOC} + \Delta\delta^{13}C_{DOC} \qquad (3)$$

$$CH_{4\text{-derived hopanoids (\%)}} = \frac{(\delta^{13}C_{hopanoid} - \delta^{13}C_{Heterotroph})}{(\delta^{13}C_{MOB} - \delta^{13}C_{Heterotroph})} \times 100 \qquad (4)$$



# 3 Results

## 3.1 Large-scale spatial patterns of dissolved CH$_4$ concentrations

The dissolved CH$_4$ concentrations in the Laptev Sea were high compared to the atmospheric equilibrium at ~4 nM. The sub-pycnocline CH$_4$ concentrations for the 26 studied stations varied widely between 8 and 11 850 nM (median = 278 (IQR = 88-

966) nM), with the highest concentrations at station 47 in the Outer Laptev Sea hotspot region (OLS; Fig.2). The sub-pycnocline CH$_4$ concentrations were significantly different between all regions (Table S1). The CH$_4$ concentrations in the OLS (defined as latitudes ≥ 76.1° N & longitudes 125-131°E) were significantly higher compared to the Inner and Mid Laptev Sea (ILS/MLS; defined as latitudes <73.9 °N & 73.9-76.1°N & longitudes ~130°E, respectively; Table S1). In all regions, dissolved CH$_4$ was significantly lower in the surface water compared to below the pycnocline (Table S2). Additionally, surface water

CH$_4$ was significantly higher in the ILS hotspot region (Table S3) compared to the MLS and OLS (Table S3).

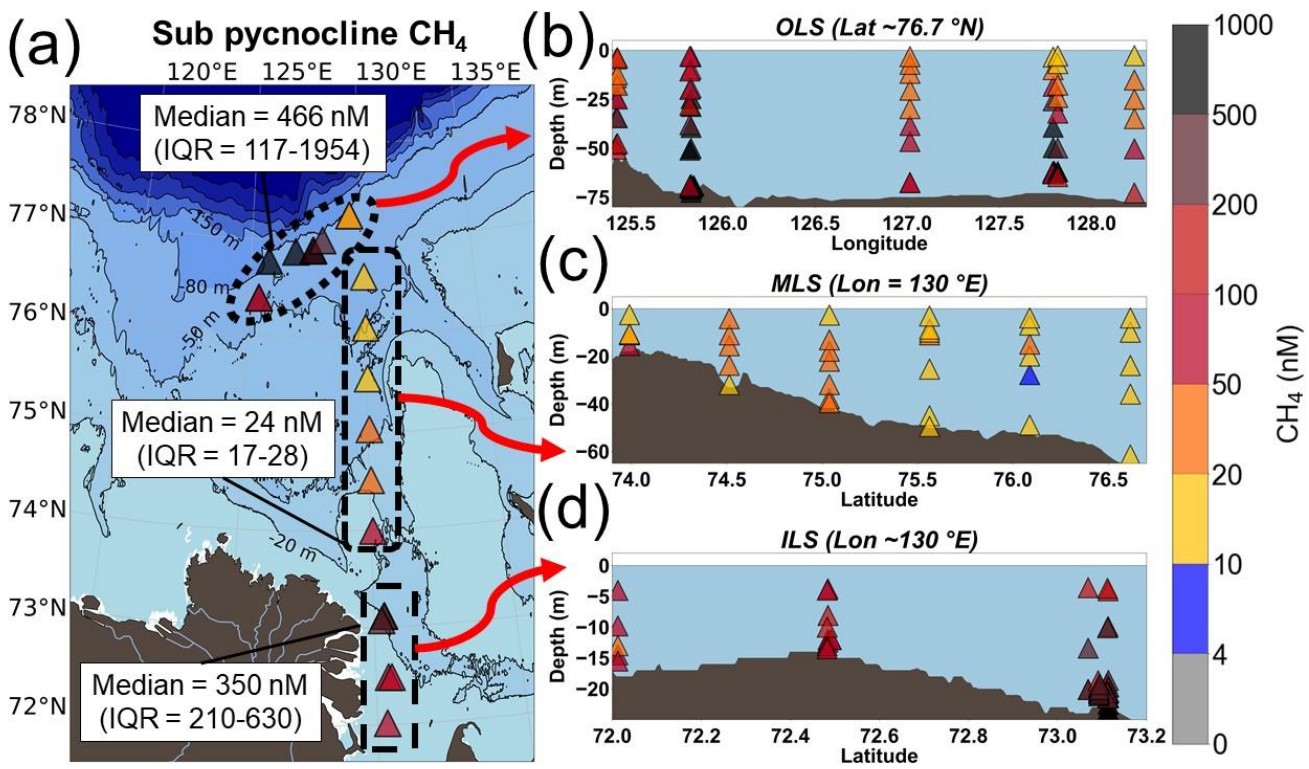

**Figure 2.** Dissolved CH$_4$ concentrations (nM). (a) Average CH$_4$ concentrations below the pycnocline. (b) The outer Laptev Sea hotspot region (OLS). (c) Mid-Outer Laptev Sea transect (MLS). (d) The inner Laptev Sea hotspot region (ILS). The bathymetry was plotted using IBCAO version 5.0 (Jakobsson et al., 2024).





### 3.2 Organic geochemistry

**3.2.1 Bulk organic carbon**

The total Organic Carbon (OC) content displayed strong variability across the Laptev Sea. The total OC varied between 0.2 and 2.3 % (mean±standard deviation; 1.0±0.5%, n=25), with the highest concentrations observed in the ILS (1.3±0.8, n=6), compared to the OLS (0.9±0.4, n=14) and MLS (1.0±0.5, n=5). The $\delta^{13}C$-OC ranged from -26.8 to -23.3 ‰ (-24.7±1.0 ‰, n=25), with the lowest values present in the ILS (-26.1±0.5, n=6) and slightly higher values in the OLS (-23.9±0.4, n=14) and MLS (-25.0±0.7, n=5).

**3.2.2 Hopanoid lipid concentrations and spatial distribution patterns**

Hopanoids were present at detectable levels across all surface sediments. The sum of $C_{30}$-hopenes ($\sum C_{30}$-hopenes; diploptene, hop-17(21)-ene and neohop-13(18)-ene) displayed a large range of concentrations across the Laptev Sea (mean±standard deviation; 27±19 µg gOC$^{-1}$; n=25). The highest concentrations were found in the OLS (35±20 µg gOC$^{-1}$; n=15; Fig.5). Similar $\sum C_{30}$-hopenes concentrations were also present in the ILS, with no significant difference compared to the OLS (18±11 µg gOC$^{-1}$; n=6; Fig.5; Supplementary Table 4). In contrast, the concentrations of $\sum C_{30}$-hopenes were significantly lower in the MLS (10±4 µg gOC$^{-1}$; n=4; Fig.5) than in the OLS, but displayed no significant difference compared to the ILS (Supplementary Table 4). On average, diploptene (hop-22(29)-ene) comprised the majority of $C_{30}$-hopenes (68%), with a lower abundance of its diagenetic products (hop-17(21)-ene and neohop-13(18)-ene; 19 & 13%). While diploptene dominated the $C_{30}$-hopene abundance across all regions, it comprised a larger part of the $C_{30}$-hopenes in the ILS (74%) compared to the MLS and OLS (52% & 69%, respectively). In the MLS in particular, hop-17(21)-ene and neohop-13(18)-ene constituted a larger part of the $\sum C_{30}$-hopenes (31 % & 18 %).

**3.3.3 Hopanoid isotope composition and spatial distribution patterns**

The compound-specific isotope composition of $\delta^{13}C$-$C_{30}$-hopenes (diploptene, hop-17(21)-ene and neohop-13(18)-ene) in the Laptev Sea was strikingly depleted compared to the bulk OC. The $\delta^{13}C$-$C_{30}$-hopenes were on average -47.8±7.2 ‰ (mean±standard deviation; n=23; Fig.5). The $\delta^{13}C$-$C_{30}$-hopenes were lowest in the OLS, with depleted $\delta^{13}C$ signals down to -57.5 ‰ at individual stations and a mean±standard deviation of -52.9±4.3 ‰ (Fig.5). Hopanoids in MLS and ILS exhibited depleted but relative to the OLS more enriched $\delta^{13}C$-$C_{30}$-hopenes (-42.0±1.9 ‰ & -39.4±3.0‰; Fig.5). No significant differences were observed between the two regions (Table S5), yet both were much more depleted than their regional $\delta^{13}C$-OC.

Amongst the $C_{30}$-hopanoids, $\delta^{13}C$-diploptene and diplopterol displayed the lowest values in the OLS (-59.5 & -61.0 ‰; Fig.3). Hop-17(21)-ene and neohop-13(18)-ene likewise demonstrated low $\delta^{13}C$, yet neohop-13(18)-ene was more enriched in $^{13}C$ in the OLS (Fig. 3). However, in the MLS/ILS diploptene and diplopterol in particular were more enriched in $\delta^{13}C$, up to a



maximum of -34.1 & -29.0 ‰, respectively (Fig.3). On the contrary, $\delta^{13}C$ of hop-17(21)-ene exhibited lower values compared to diploptene in the ILS (-41.9±4.6 ‰ & -37.6±2.2 ‰; Fig.3). While diplopterol was also present in the ILS samples, coelution with other molecules hindered accurate concentration and isotope measurements due to interferences. Hence, $\delta^{13}C$-diplopterol was not quantified in the ILS. In summary, $\delta^{13}C$-$C_{30}$-hopanoids were generally low across the Laptev Sea, especially in the

OLS. However, in the ILS, hop-17(21)-ene was more depleted in comparison to other hopanoids (Fig.3).

**Figure 3.** (a) Overview of the measured $\delta^{13}C$ of individual $C_{30}$ hopanoids across each region. The shaded gray zone indicates the range of $\delta^{13}C$-heterotrophic hopanoids by production from DOC after isotope fractionation during hopanoid synthesis (Supplementary Tables S6 & S7; Salvado et al., 2016; Schwartz et al., 2023). The light blue indicates the measured $\delta^{13}C$-OC in this study. (b) Schematic of $\delta^{13}C$ of carbon
substrates used for hopanoid production and the associated isotope fractionation ($\Delta\delta^{13}C$; Supplementary Tables S6 & S7). Methane Oxidizing Bacteria (MOB; orange) in each region are the $\delta^{13}C$-hopanoid endmembers, where the isotope fractionation of MOB-I and II, respectively,





was weighted against the relative fraction of MOB (Eq. 2). The pink displays the $\delta^{13}C\text{-}CH_4$ in each investigated region (Supplementary Table S6). (c) Measured $\delta^{13}C\text{-}C_{30}$ hopanoids compared against the hopanoid and carbon source endmembers in each region.

### 3.3 Bacterial consortia of hopanoid producers

Aerobic methanotrophs, here referred to as Methane Oxidizing Bacteria (MOB), were present in 9 out of the 11 sediment samples probed for 16S-rRNA. Both type I (*Gammaproteobacteria*; MOB-I) and type II (*Alphaproteobacteria*; MOB-II, Fig.4; Fig. S1) MOB were detected in those 9 samples. Type I MOB constituted the majority of the MOB in the OLS compared to MOB-II (mean±standard deviation;1.2±1.8 % of all sequences, n =5; 0.10±0.08, n=3; Fig.4). Specifically, the highest relative abundance of MOB-I was present at station 47 (4.3 %: Fig.3). In the ILS MOB I and II were of similar relative abundances

(0.18±0.15 %, n =2 & 0.17±0.15 %, n=3), yet almost a factor ten lower than in the OLS. While the relative abundances of MOB were overall low in the MLS, MOB-II were more abundant (0.04±0.01 n = 2 & 0.01±0.01, n=2; Fig. 4). The most common MOB-I were of the order *Methylococcales*, specifically of the genera *Methyloprofundus* and *Marine methylotrophic group 2* (Fig. S1). Other common type I MOB in the Laptev Sea were *pLW-20, Milano-WF1B-03, Milano-WF1B-42* and *pItb-vmat-59* (Fig. S1). *Crenothrix* was only detected in the ILS (Fig. 4; Fig. S1) and *IheB2-23* and *Cycloclasticus* only at the OLS

stations (Supplementary Fig.1). Type II MOB were of the family *Methyloligellaceae*¸ with two identified genera (*Methyloceanibacter & Methyloligella*). In addition to MOB, heterotrophic and autotrophic hopanoid producers were present in the Laptev Sea, particularly in the ILS (Fig. S2). Taken together, MOB-I constituted the largest part of the bacterial hopanoid producers in the OLS, with almost ten times higher relative abundance compared to the MLS/ILS. Hopanoid producers in the ILS/MLS were a mix of MOB-I/II and non-methanotrophic hopanoid producers, primarily heterotrophic bacteria (Fig.4).

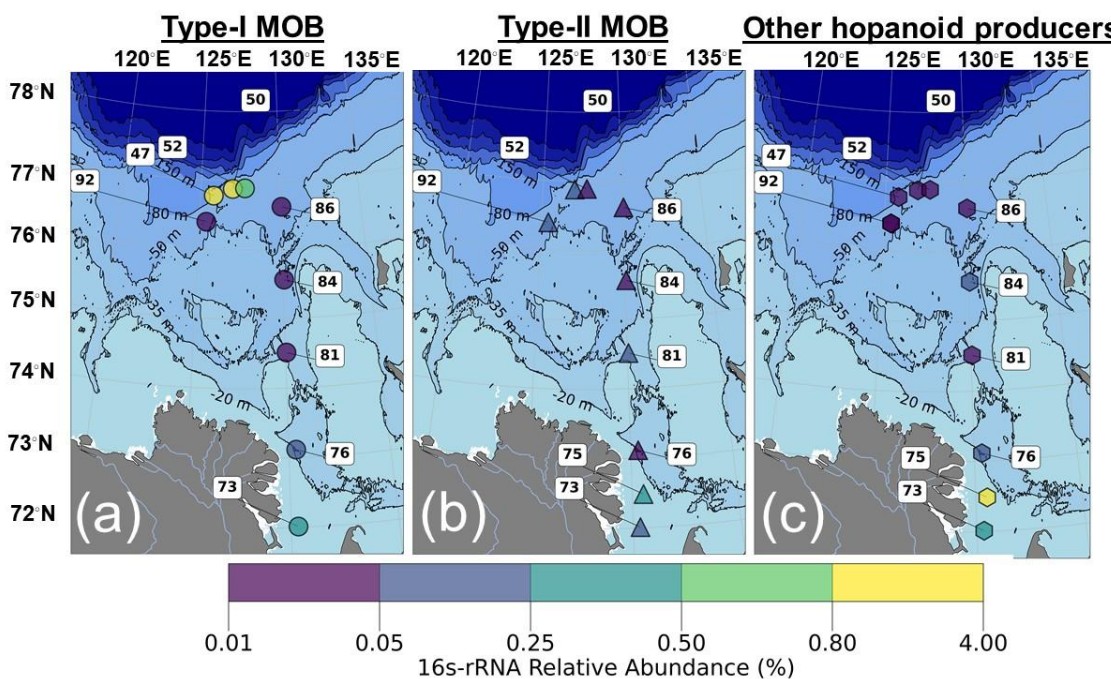




**Figure 4.** Relative abundance of hopanoid producers in the surface sediments (%), based on 16S-rRNA. (a) Type I MOB. (b) Type II MOB. (c) Other hopanoid producers. The bathymetry was plotted using IBCAO version 5.0 (Jakobsson et al., 2024).

## 4 Discussion

### 4.1 Widespread occurrence of lipid biomarkers tracing methane release across the Laptev Sea

Depleted $\delta^{13}$C-C$_{30}$ hopanoids in surface sediments indicate large-scale CH$_4$ release across the Laptev Sea. As the Laptev Sea water and surface sediments are well oxygenated (Stepanova et al., 2017; Brüchert et al., 2018; Maciute et al., 2025), the CH$_4$ signal in hopanoids is most likely related to aerobic processes. The Laptev Sea is frequently ventilated through storms, resulting in a wide variety of dissolved CH$_4$ concentrations dependent on the time of sampling relative to storm ventilation of the water column CH$_4$ (e.g., Shakhova et al., 2014). In contrast, the CH$_4$ signal from $\delta^{13}$C-C$_{30}$ hopanoids accumulated in sediments

indicates a CH$_4$ release signal averaged over several years. Thus, $\delta^{13}$C-C$_{30}$ hopanoids provide means to assess long-term CH$_4$ release patterns compared to the ephemeral dissolved CH$_4$ concentrations in the Laptev Sea. Hence, the $\delta^{13}$C-C$_{30}$ hopanoids help to unearth CH$_4$ dynamics in a region that is hard to access and thereby void of time-series monitoring of CH$_4$. The proxy-derived CH$_4$ signal in the isotopes of these hopanoids also corroborates previous suggestions of the Laptev Sea as a region of widespread CH$_4$ cycling in the water column (e.g., Semiletov, 1999; Shakhova et al., 2010, 2013, 2014; Steinbach et al., 2021).

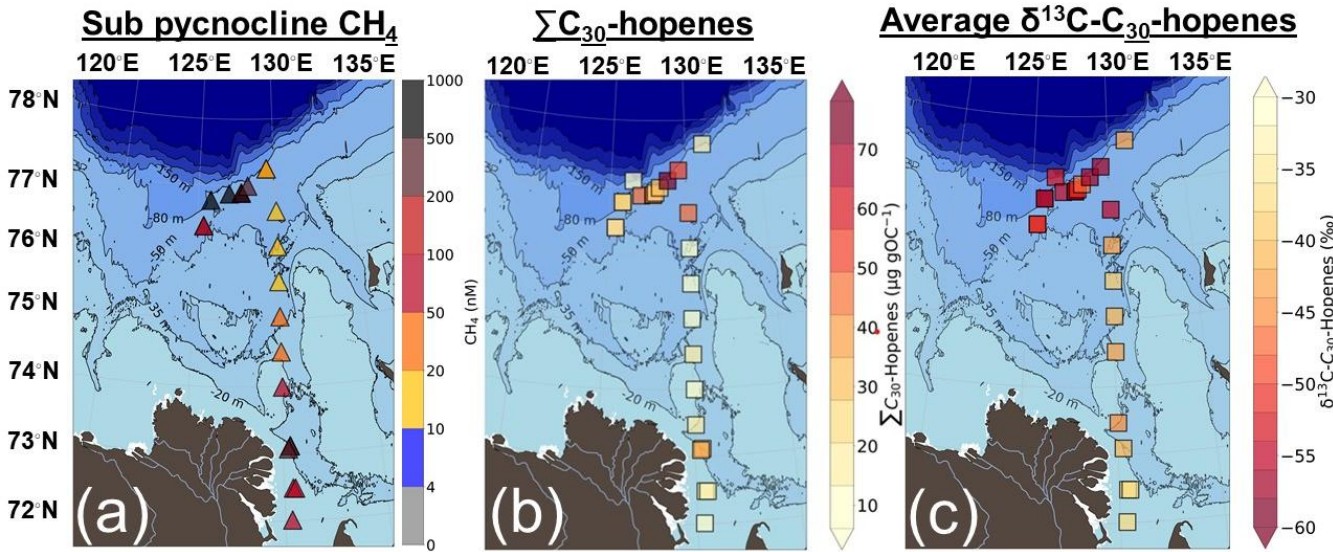


**Figure 5.** (a) Sub pycnocline CH$_4$ concentrations, (b) the total concentration of C$_{30}$ hopenes (diploptene, hop-17(21)-ene & neohop-13(18)-ene; µg gOC$^{-1}$) and (c) the average $\delta^{13}$C-C$_{30}$ hopenes (‰) in Laptev Sea surface sediments. The bathymetry was plotted using IBCAO version 5.0 (Jakobsson et al., 2024).



### 4.2 Spatial variations of methane concentrations and hopanoid-isotope patterns

#### 4.2.1 Intensive $CH_4$ cycling in the Outer Laptev Sea hotspot region

The high abundance of very depleted $\delta^{13}C$-$C_{30}$ hopanoids in the OLS highlights ubiquitous release of $CH_4$ with subsequent AeOM across this region (Fig.5). High concentrations of $C_{30}$ hopenes in the OLS hotspot region are broadly consistent with previous observations of diploptene from surface sediments sampled in this area in 2011 and 2016 (Grinko et al., 2020, 2025). Beyond earlier studies, our findings of depleted $\delta^{13}C$-$C_{30}$ hopanoids in this region confirm a methanotrophic source. The presence of very low $\delta^{13}C$-$C_{30}$ hopanoids in the OLS is a strong indication of MOB-I, which is further corroborated by 16S-rRNA (Fig. 4 & 5). Indeed, the application of a stable isotope mixing model (Eq. 4) suggests that 72±6 % of hopanoids in the OLS are related to MOB-I and II. As the mixing model is weighted against the abundance of MOB-I and II and not their respective activity, these results should be regarded as semi-quantitative. Nonetheless, it supports the hypothesis of $CH_4$ incorporation in hopanoids. Additionally, diplopterol is slightly more depleted in $^{13}C$ in the OLS compared to other $C_{30}$ hopanoids; this might reflect increased depletion in $^{13}C$ of hopanoids for each step in the biosynthetic pathway (i.e., $^{13}C$ isotope depletion from squalene to diploptene, to diplopterol; Summons et al., 1994). In short, very low values of $\delta^{13}C$-$C_{30}$ hopanoids together with high $CH_4$ concentrations and a high relative abundance of MOB-I indicate the OLS as a region of extensive $CH_4$ emissions and associated AeOM (Fig. 2b, 5).

#### 4.2.2 Varying hopanoid sources in the Inner Laptev Sea hotspot region

High concentrations of $CH_4$, depleted $\delta^{13}C$-$C_{30}$ hopenes values and the presence of MOB highlight the possibility of tracing $CH_4$ release also in the ILS. However, while concentrations of $C_{30}$ hopenes were relatively high and not significantly different from those in the OLS, the $\delta^{13}C$-$C_{30}$ hopenes in the ILS are less depleted. This can be interpreted in different ways, but is likely a combination of three mechanisms: (1) the relative contribution from MOB-I and II, (2) isotopic dilution from non-methanotrophic sources and (3) lower activity of AeOM in the ILS compared to the OLS. In contrast to the OLS, a higher relative abundance of MOB-II is present in relation to MOB-I in the ILS. Type II MOB can produce hopanoids more enriched in $^{13}C$ compared to MOB-I due to their ability to also assimilate $CO_2$ into biomass (Jahnke et al., 1999). *Methyloceanibacter* constituted the majority of MOB-II in the ILS (Fig. S1) and is common in hydrothermal vents in marine systems (Takeuchi et al., 2014). Thus, the majority of the MOB-II in our study is likely an in situ signal of AeOM rather than of coastal influence as was found in the Kara Sea/Yenisei River (de Jonge et al., 2016), despite MOB-II generally dominating terrestrial ecosystems (Hanson & Hanson, 1996; Inglis et al., 2019). The less depleted $\delta^{13}C$-$C_{30}$ hopenes, yet in similar concentrations to the OLS, can therefore be an indication of MOB-II and a proxy-derived $CH_4$ signal in coastal regions needs thorough system knowledge to depict the source.

The higher $\delta^{13}C$-$C_{30}$ hopenes in the ILS compared to the OLS can also be explained by a dilution of hopanoid production from non-methanotrophic bacteria. Hopanoids (e.g., diploptene, hop-17(21)-ene) are directly synthesized or produced as



degradation products of bacteriohopanepolyols (BHPs; Rohmer et al., 1980; Mackenzie et al., 1981). A mixture of hopanoid sources is to be expected, as ~10 % percent of bacteria can synthesize hopanoids (Belin et al., 2018). Relatively high concentrations of bacteriohopanetetrol (BHT) have previously been found in the ILS and have been attributed to terrestrial sources (Bischoff et al., 2016). This study thereby indicates that a dilution from non-methanotrophic terrestrial sources may

contribute to the higher $\delta^{13}C$-$C_{30}$ hopenes, due to the vicinity of the Lena River delta and higher relative abundance of non-methanotrophic hopanoid producers compared to MOB. Hopanoid source apportionment displayed that 78±7 % of hopanoids in the ILS are related to non-methanotrophic hopanoid synthesis. Consequently, the presence of hopenes more enriched in $^{13}C$ is also an indication of an additional contribution from non-methanotrophic hopanoid sources, rather than reflecting lower $CH_4$ releases from this region.


Despite high $CH_4$ concentrations in the ILS, higher $\delta^{13}C$-$C_{30}$ hopenes compared to the OLS hotspot region could alternatively also indicate less active AeOM. Incubation experiments of ILS surface sediments and water show the possibility of AeOM (Bussmann et al., 2017, 2021; Tikhonova et al., 2021), yet at lower rates compared to the OLS (Tikhonova et al., 2021). Taken all together, the relative enrichment of $\delta^{13}C$-hopenes in the ILS compared to the OLS hotspot is likely a combination of: (1) a

higher relative input of hopanoids from MOB-II, (2) non-methanotrophic bacteria produced in situ and in terrestrial settings and (3) lower activity of AeOM compared to the OLS. Nonetheless, the depleted $\delta^{13}C$-hop-17(21)-ene does indicate the presence of AeOM and thereby $CH_4$ release (Fig.3).

### 4.2.3 AeOM-biomarkers tracing methane release in the Mid-Outer Laptev Sea transect

In the MLS, lower yet readily detectable $C_{30}$ hopene concentrations suggest less MOB biomass present in this region.

Nonetheless, the depleted $\delta^{13}C$-$C_{30}$ hopenes signals and presence of MOB in the MLS regime indicate hopanoids tracing $CH_4$ release (Fig. 4 & 5), with the source apportionment indicating that MOB contributed to 58±15 % of the hopanoid production. In situ hopanoid production in the MLS region or surface water transport of hopanoids from the ILS hotspot are possible causes. Given that $CH_4$ concentrations are higher in the sub-pycnocline waters of the MLS than in the surface water, where transport from the ILS would be centered and that a larger fraction of hopanoids comes from MOB compared to in the ILS, it

seems likely that most hopanoids in the MLS are locally produced. Prior assessments have considered the MLS as a region of continuous subsea permafrost and comparatively low $CH_4$ concentrations (Shakhova et al., 2010; Bukhanov et al., 2023). In contrast, we here show a proxy-derived record of $CH_4$ in a region that was previously conceived as being of low $CH_4$ release. However, additional modern calibrations of these hopanoid proxy tools in regions with lower $CH_4$ concentrations are necessary to confirm whether this indicates an onset of $CH_4$ release, or if it is regarded as a "background signal" of $CH_4$-related hopanoids

in marine systems. Taken together, we display a record of biomarker proxy-derived $CH_4$ in the MLS, most likely dominated by a more diffuse regional $CH_4$ source in the MLS. The presence of low $\delta^{13}C$-$C_{30}$ hopenes across the Laptev Sea indicates the potential to trace proxy-derived $CH_4$ in all investigated regions and displays important factors to consider when interpreting such biomarker records.





**4.3 Large-scale implications for Laptev Sea CH$_4$ cycling**

Since the discovery two decades ago, the Laptev Sea CH$_4$ dynamics and sources have been increasing. It has been suggested that gas-hydrate-bearing sediments in the region store vast amounts of CH$_4$ (Soloviev, 2000), functioning as a possible CH$_4$ source upon destabilization (e.g., Shakhova et al., 2010, 2011). Thawing subsea permafrost has also been suggested as a source of CH$_4$ in the ILS (Shakhova et al., 2014; Sapart et al., 2017) and to serve as a conduit for leakage of deep thermogenic CH$_4$ gas reservoirs in the OLS (Steinbach et al., 2021). Our display of AeOM across the Laptev Sea indicates the presence of a

microbial sink, but the high CH$_4$ concentrations question its efficiency. Historically, a large focus has been given to the ILS and OLS regions with detected CH$_4$ ebullition (e.g., Shakhova et al., 2010, 2014; Sapart et al., 2017; Steinbach et al., 2021), but few investigations have paid attention to the MLS as the observations have reported lower CH$_4$ concentrations in that region, yet still elevated relative to atmospheric partial pressures. Thus, our indications of CH$_4$ release in the MLS stimulate a few questions. (1) Why has subsea permafrost-related CH$_4$ release only been observed in the ILS/OLS? (2) Is the CH$_4$ activity

in the MLS linked to ILS CH$_4$ release, or is there a large-scale diffuse source across the MLS?

Many challenges still exist regarding our understanding of the state of current subsea permafrost and associated CH$_4$ releases. In situ bottom sediment temperatures point towards thawing permafrost in the ILS/OLS CH$_4$ hotspot regions (Chuvillin et al., 2022; Bukhanov et al., 2023). However, the state of the subsea permafrost and consequent CH$_4$ releases from these two regions

are likely caused by different mechanisms. Inundation of the OLS and accompanied heat exposure from both Atlantic water and underlying fault zones transporting geothermal heat have likely contributed to its warming thermal state (Chuvillin et al., 2022). Indeed, areas of warmer sediment temperatures in the OLS are coinciding with CH$_4$ seepages and underlying fault zones (Baranov et al., 2020; Chuvillin et al., 2022). In the ILS, heat flux from the Lena River is proposed to be the main mechanism thawing subsea permafrost, but thawing is also likely induced by geothermal heat through the Ust' Lenskii Rift (Chuvillin et

al., 2022; Bukhanov et al., 2023). Additionally, the bottom water from the Lena River plume has increased by ~1°C between 1999-2012 (Shakhova et al., 2014) during which time subsea permafrost has rapidly thawed (Shakhova et al., 2017). Thus, the state of subsea permafrost and lower CH$_4$ release in the MLS can be related to the mechanisms regulating the thermal state of subsea permafrost (i.e., heat flux from geothermal or riverine sources).

The surface water CH$_4$ concentrations in the MLS are lower than in the ILS and OLS, yet they exceed the atmospheric equilibrium (4 nM) by ~4.6 times (median = 18.6 nM). However, the source of CH$_4$ in the MLS is not fully clear. Assuming that a regional CH$_4$ source is present in the MLS, further investigations are necessary, especially if it is related to the subsea permafrost thaw-state. In situ bottom sediment temperatures indicate that the MLS is on the border of discontinuous/continuous subsea permafrost (Chuvillin et al., 2022; Bukhanov et al., 2023) and thus possibly on the "thaw-front", which could be an

indication of the thaw-state regulating the CH$_4$ emissions from the MLS. If a regional-MLS source of CH$_4$ is present in this region, the source of these CH$_4$ emissions and the associated AeOM needs to be further investigated. Taken together, there





may be geospatial thaw fronts of the subsea permafrost from the OLS moving inwards and from the ILS moving outwards, thus both over time leading to a shrinking MLS transition zone and growing coverage of the extensive areas with accompanying $CH_4$ releases.

## 5 Conclusions

This study demonstrates that $\delta^{13}$C-$C_{30}$ hopanoids can be used as time-integrated proxies to deduce geospatial patterns of $CH_4$ release from marine sediments. A clear signal of $CH_4$ being incorporated into hopanoids is shown throughout the wide outer Laptev Sea hotspot region, indicating the possibility to also investigate time trends of $CH_4$ release from this region in chronologically-constrained sediment cores. Additionally, a $CH_4$ release signal is discovered in the Mid-Outer Laptev Sea transect (MLS), previously thought to be a region of comparatively low $CH_4$ cycling. The MLS signal could be derived from the Inner Laptev Sea hotspot (ILS), yet, more likely, from a diffuse source in the MLS itself. In the ILS, high $CH_4$ concentrations were less well reflected in the molecular fossils, indicating AeOM, yet still indicative of $CH_4$. This is attributed to a higher relative abundance of MOB-II that utilize both $CH_4$ and $CO_2$ for hopanoid production, isotopic mixing with non-methanotrophic bacteria producing hopanoids, but also less active AeOM in the ILS hotspot. Low AeOM in a region of high $CH_4$ concentrations, such as the ILS, could have consequences for the amount of $CH_4$ emitted to the atmosphere. To the best of our knowledge, this is the first modern calibration of $\delta^{13}$C-hopanoids over an extensive marine setting, facilitating improved interpretations of hopanoid sources in contemporary and past climates. Furthermore, this study provides a large-scale geospatial picture of Laptev Sea $CH_4$ cycling, with considerations for future studies of $CH_4$ release in the circum-Arctic shelf regions.

## Data availability

The supplementary information (pdf-file) contains the composition of hopanoid producers, results from statistical analyses, chromatograms, and mass spectra of biomarkers. The supplementary data can be accessed through the Bolin Centre Database (link to be published).

## Author contributions

This study was conceptualized by Ö.G and A.E. Shipborne expedition and at-sea sampling strategies were conceived by I.S, Ö.G., N.S and B.W. Shipborne sampling and protocols were performed by Ö.G, I.S, B.W, D.K, H.H. Laboratory procedures, calculations, mapping and the first draft of the manuscript were made by A.E., in collaboration and under supervision of Ö.G., B.W., W-L.H., and H.H. All the authors contributed to data interpretation and writing.



**Competing interests**

The authors declare no conflicts of interest.

**Acknowledgements**

We thank Dr. Jannik Martens, Dr. Tommaso Tesi, and the rest of the ISSS-2020 participants, enabling the expedition, including sediment and CH$_4$ sampling. Additionally, we thank Dr. Qingzeng Zhu, Dr. Xinwei Yan, Prof. Weichao Wu, and Felipe Matsubara for helpful discussions regarding laboratory protocols.

**Financial support**

This research was funded by the Swedish Research Council (Grant 2017-01601 to Ö.G; FORMAS grant 2021-01750 to B.W), Knut and Alice Wallenberg Foundation (Wallenberg Academy Fellowship 2023.0058 to B.W, and Tipping Arctic Ocean Methane Project, grant KAW 2024.0140 to Ö.G), and the European Research Council (ERC Advanced Grant CC-TOP 695331 to Ö.G). The charter of the RV Academician Mstislav Keldysh was funded by grant 121021500057–4 to POI from the Ministry

of Science and Higher Education of the Russian Federation, and the Russian Scientific Foundation (grant 21-77– 1243 30001 to IS; grant 22-67–00025 to NS).

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
