# Peer review of "Methane releases across the Laptev Sea signaled by time-integrated biomarkers of aerobic methane oxidation"

_EGUsphere, 2025_

## Author Comment (AC1)

**Author response to referee comments and the resulting revisions of ms egusphere-2025-4756: "Methane releases across the Laptev Sea signaled by time-integrated biomarkers of aerobic methane oxidation"**

Reference: [https://doi.org/10.5194/egusphere-2025-4756](https://doi.org/10.5194/egusphere-2025-4756)

Albin Eriksson, Birgit Wild, Wei-Li Hong, Henry Holmstrand, Francisco Jardim de Almada Nascimento, Stefano Bonaglia, Denis Kosmach, Igor Semiletov, Natalia Shakhova, and Örjan Gustafsson.

We gratefully thank the referees for thoroughly reviewing and giving constructive feedback that helped to clarify the significance and importance of this manuscript during revision.

All reviewer comments are included below in *black italic font* each followed by our detailed author responses, formatted as indented blue text. Citations of our implemented changes in the manuscript are formatted as *indented italic blue text*.

*Anonymous Referee #1*: https://doi.org/10.5194/egusphere-2025-4756-RC1

*This study utilizes lipid biomarker signatures from surface sediment samples to obtain a time-integrated signal of methane release events in the Laptev Sea. The results suggest that methane oxidation occurs across the Laptev Sea shelf, including the mid-shelf region that was previously known as a region of low methane emissions. This study is relevant for the community, and for better understanding methane cycling in the Arctic. Overall, it is very well-written and very thorough. I have several minor comments and suggestions regarding the manuscript.*

> We are grateful for the supportive comments about the importance of this work for the community and how it improves the knowledge about Arctic methane cycling. Referee #1 provides below great suggestions about the possible hopanoid producers, including the possibility to expand that list to further improve the biomarker interpretation. We thank referee #1 for the suggested changes which positively influenced our revisions and helped improve the quality of the manuscript further.

*Comments:*

*Lines 23-24, 76, and 81: The authors interpret the d13C values of the hopanoids as a proxy for "methane release". In principle, yes, there does appear to be a correlation between methane concentrations and d13C values, however, this has not been thoroughly tested in diverse environmental settings and with different methanotroph communities. It would be more accurate to say that this proxy reflects aerobic methane oxidation, and therefore enhanced methane cycling. I would recommend rephrasing this in both the abstract and the manuscript by replacing "methane release" with "enhanced methane cycling" or "methane oxidation".*

> We agree with this distinction. "Methane release" has been rephrased as "enhanced methane cycling" throughout the manuscript.

*Lines 86-87: Based on previous studies, does methane-oxidation predominantly occur in the sediment or the water column? Or does this vary based on the site?*

> To our current understanding, aerobic methane oxidation (AeOM) in this region has been observed in near bottom/sub-pycnocline waters of the outer Laptev Sea (Shakhova et al., 2015; Samylina et al., 2021) and in the inner Laptev Sea (Bussmann et al., 2021). Observations of AeOM and AOM has also been seen in Outer Laptev Sea sediments (Tikhonova et al., 2021; Savvichev et al., 2023). Based on these studies, with limited information on incubation-based rates of methane oxidation (both AeOM and AOM) and whether it mostly occurs in the water or sediments are variable between their studied stations. Similar studies remain to be made in other parts of the Laptev Sea to fully understand which process is of highest importance.

> Our study recognizes that both AeOM in the water column and the oxygenated surface sediments are possible, but cannot distinguish between the importance of the two with our biomarker approach. Both possibilities are already mentioned in the submitted ms on lines 86-87 and 402-403.

*Lines 268-275: Based on these measurements the methane concentrations are highly variable between sites as you mentioned. Is most of the methane being emitted from the Laptev Sea from a diffusive or ebullitive flux? Previous studies have shown that aerobic methanotrophs are usually less efficient with oxidizing methane when the methane flux is mostly ebullitive. If the flux is mostly ebullitive rather than diffusive, do you expect this to influence the proxy signatures and your interpretations? Coming back to my previous comment, is the water fully oxygenated at all these sites? If so, do you expect most of the methane-oxidation to occur in the water column and sediment, and how would this influence the proxy signatures you obtain?*

We thank referee #1 for suggesting a clarification of the importance of the ebullitive versus diffusive fluxes of methane. Shakhova et al. (2010, *Science*) estimated that ~40 % of the methane fluxes to the atmosphere are from diffusive fluxes, and the rest from ebullitive fluxes. In our study, there are two main regions where ebullition is widespread, the OLS and the ILS. As evident from the very depleted $\delta^{13}$C-hopanoids in the OLS (52.9±4.3 ‰, n =14) methane is incorporated into the hopanoid biomarkers across the samples of that region. Notably, the data also display strong evidence of AeOM where ebullitive fluxes dominate; while dissolved methane concentrations are variable, the median is still also in these bubble regimes very high at 350 nM in the ILS and 466 nM in the OLS (Fig.2), supporting that lots of dissolved methane is also here available for oxidation. Therefore, it likely does not affect the proxy-signature.

Regarding the oxygenation of the water column/sediments, multi-year observations of oxygen (2015-2020) in the Laptev Sea water column displayed a usual oxygen saturation between 70-100 % (Xie et al., 2023, *Front. Mar. Sci.*). Sediments from the same stations that were sampled in our study displayed oxygen penetration depths between 0.2-1.8 cm (Maciute et al., 2025, *Env. DNA*). Taken together, this displays the possibility for AeOM throughout the oxygenated water column and in the oxygenated surface sediments, which has also been observed in previous studies (Shakhova et al., 2015, *Philos. Trans. R. Soc. A.*; Tikhonova et al., 2021, *Microbiology*; Bussmann et al., 2021, *Biogeosciences*). However, whether most of the oxidation occurs in the surface sediments or in the water column still remains unresolved, yet is of lesser relevance for the present study.

*Lines 335-336: Not all members of the family Methyloligellaceae are considered methanotrophs. The majority of the classified strains appear to be methylotrophs, including Methyloligella and certain strains of Methyloceanibacter. You can still mention these as potential candidates for Type II methanotrophs, but you should add a sentence somewhere indicating that these are not all necessarily methanotrophs. Further, do you know whether Methyloligella and Methyloceanibacter have the capacity to produce hopanoids? It would be good to confirm this as this can influence some of your interpretations. You can check this by searching for the sqhc gene (accession no. WP_038942977.1) on the NCBI database, and then checking to see if either of these species contain the gene (see Richter et al. 2023 Biogeosciences for more details).*

We thank referee #1 for pointing out that not all *Methyloligella* and *Methyloceanibacter* are methanotrophs. We have consequently change lines 335-336 to:

*"Candidates of Type II MOB were of the family Methyloligellaceae¸ with one identified genus known to produce hopanoids (Methyloceanibacter). However, it should be recognized that not all Methyloligellaceae are necessarily methanotrophs"*.

We also thank referee #1 for the very helpful suggestion to check whether *Methyloligella* and *Methyloceanibacter* produce hopanoids through accession no. WP_038942977.1. From this search three *Methyloceanibacter* species were found (*Methyloceanibacter methanicus*, *Methyloceanibacter ceanitepidi* and *Methyloceanibacter stevinii*). However, no *Methyloligella* species encoding *shc* (squalene-hopene cyclase) were found.

As a result of no known *Methyloligella* producing hopanoids, we have limited the MOB-II to *Methyloceanibacter* and revised the related results (isotopic mass balances, relative abundances of MOB-II, figure S1, figure 3 and figure 4).

*Lines 344-354: The title of this subsection and the content seem unrelated. This section also sounds like it belongs more in the conclusions rather than at the start of the discussion.*

The incorporation of this section and header is to highlight that enhanced methane cycling was displayed across the Laptev Sea in the lipid biomarkers even though there were regional differences. This is an important section to clarify that the proxy works in the Laptev Sea before discussing the details of each region. To clarify this the section heading 4.1 has been changed to:

*"Indications of widespread enhanced methane cycling indicated by lipid biomarkers across the Laptev Sea"*

*Line 381-383: Are there hydrothermal vents in the ILS region?*

Thermospores of hydrothermal origin have been observed in the outer Laptev Sea (Ståhl et al., 2024, *Geobiology*), but not in the Inner Laptev Sea.

To avoid confusion regarding hydrothermal vents, we have rephrased the sentence in line 381-383:

*"Methyloceanibacter constituted the only genera of MOB-II in the ILS (Fig. S1) and has to this date to our knowledge only been isolated from marine systems (Takeuchi et al., 2014, Int. Journ. System. Evol. Biol.; Takeuchi et al., 2019, PLoS ONE; Vekeman et al., 2016, Environmental Microbiology). "*

*Lines 382-383: The reference that you site here (Takeuchi et al. 2014) says the strain of Methyloceanibacter is a methylotroph and not a methanotroph. Going back to my previous comment, maybe check to see if they produce hopanoids.*

We thank referee #1 for the suggestion to search for the possibility of methane-related hopanoid production from *Methyloceanibacter*. As noted from the previous comment,

*Methyloceanibacter methanicus*, *Methyloceanibacter ceanitepidi and Methyloceanibacter stevinii* are capable of producing hopanoids, with *Methyloceanibacter methanicus* being a methanotroph (Vekeman et al., 2016, *Environmental Microbiology*). Therefore, *Methyloceanibacter* can be methanotrophs, but not necessarily. As referee #1 suggested earlier, we have now clarified this in lines 335-336 (see our previous comment).

*Lines 383-387: Since it is unclear whether some of the MOB you have classified as Type II MOB are methanotrophs, I would say it is difficult to fully exclude terrestrial inputs in the ILS region. Do you have any other independent biomarkers that can give an indication of how much of your signal here is derived from terrestrial sources? In your next paragraph, you seem to indicate that terrestrial inputs are relatively high, so you might still have a terrestrial signal from your methanotrophs.*

We thank referee #1 for the suggestion to clarify the source of the hopanoid signals of the ILS. Given the abundance also of "other hopanoid producers" in figure 4, we cannot fully exclude a terrestrial influence for the hopanoids. However, the MOB-II present in the ILS (*Methyloceanibacter;* Fig S1*)* have to our knowledge only been isolated from marine environments (Takeuchi et al., 2014, *Int. Journ. System. Evol. Biol.; Takeuchi et al.,* 2019, *PLoS ONE*; Vekeman et al., 2016, *Environmental Microbiology*). Additionally, the MOB-I in the ILS (*Marine methylotrophic group 2*, *Methyloprofundus*, *Milano-WF1B-03* and *pLW-20*) are associated with marine environments (e.g., Tavormina et al., 2015, *Int. Journ. System. Evol. Biol*; de Groot et al., 2025, *Biogeosciences*), although there is one report of *Methylosoma* having been observed in freshwater systems (Rahalkar et al., 2007, *Int J Syst Evol Microbiol*). Therefore, we conclude that the methane related signal in hopanoids of the ILS likely is predominantly from in situ production because of the presence of MOB associated with marine environments (Fig S1). However, "other hopanoid producers" diluting the methane related signal such as *Burkholderia*, *Jatrophihabitans, Bryobacter* and *Bradyrhizobium* are likely of terrestrial origin.

Regarding the methane in the ILS (likely stimulating the presence of methanotrophs), it is quite clear from the geochemistry that it is overwhelmingly stemming from coastal sediments as there are very high concentration gradients, intensive bubbling, and much lower concentration in the river waters (e.g., Shakhova et al., 2007, *J. Mar. Sys*; Shakhova et al., 2010, *Science; Shakhova et al.,* 2014, *Nat. Geo.*; Shakhova et al., 2017, *Nat. Comm*).

*Line 392: Check your reference for the "10% bacteria". Belin et al (2018) was not the first paper to report this, this was already shown in previous studies.*

We thank referee #1 for the presence of earlier literature on this, and to not only cite the review article. The sentence has been changed to "A mixture of hopanoid sources is to be expected, as ~10 % percent of bacteria can synthesize hopanoids (Ourisson et al., 1979, *Pure Appl. Chem*; Fischer et al., 2005, *Geobiology*; Racolta et al., 2012, *Proteins*; Belin et al., 2018, *Nat. Rev. Micro.*)."

*Line 401: Could the high methane concentrations in the ILS region also be derived from the Lena River rather than in situ production in the ILS, or a combination of both? Based on*

*your figure 3, it seems like the methane in this region should be very depleted. Could this tell us a bit more about the source of the methane in this region?*

> We thank referee #1 for suggesting a clarification of what the methane source is in the ILS. As discussed above, Shakhova et al. (2010, *Science*) and others showed decreasing dissolved methane concentrations through the main outflow from the Lena River, and that the methane concentrations increased in coastal waters of the ILS. In fact, the concentrations are strongly elevated in the coastal waters relative to in the river water, hence why the source must be in the coastal marine system where there is widespread ebullition observed. Additionally, isotopic evidence points towards an old biogenic source in this region, very likely from subsea permafrost (Sapart et al., 2017, *Biogeosciences*).

*Line 417: change to "here we show"*

> This sentence has been changed to "*here we show*" (line 417).

*Line 429: Change "Our display" to "Our biomarkers"*

> This sentence has been changed to "*Our biomarkers*" (line 429).

*Comments on figures:*

*Figure 1: The numbers for the stations are hard to read in the figure and against the subsea permafrost shading. Consider making the numbers black to make them easier to read. It would also be helpful if you could indicate the outer, mid-, and inner Laptev Sea regions in this figure.*

> Thank you for these helpful suggestions. The outer, mid- and inner Laptev Sea regions have been added to display the different study regions according to the referee suggestion. The station numbers were initially changed to filled black numbers according to the referee suggestion. However, this made station "45" and "75" harder to read. Therefore, the station number colors were changed to yellow with a black outline to enable a clearer distinction to the subsea permafrost lines in the background. Please see draft of revised Fig below.

[Figure]

*Figure 3: In the caption you say the shaded "gray zone" but in the figure it looks "green" to me. Consider changing this. All of these figures are showing the same things but the varying scales are a bit confusing. Consider making this into one large figure that contains all of the same information to make it easier to read.*

The text in the Fig 3 caption "shaded gray zone" has been changed to the correct color "*shaded green zone*". We are unsure what the referee is suggesting with "one large figure". We attempted to make the figure with one subplot, but the result contained to many boxes, arrows etc., making the result hard to interpret. Therefore, we have kept the figure as three subplots with one displaying the overview (a), the second the different isotopic endmembers and included isotope fractionation (b), and the third our data plotted on top of the endmembers (c). The reason behind the varying x-axis scales is to make it easier to visualize details of the results in the overview (a) panel.

*Figures 4 & 5: It would be helpful if you could indicate the ILS, MLS, and OLS regions on these figures. It would make the figures easier to interpret and to know which station numbers and data points belong to which region.*

Figure 4 and 5 have been changed according to referee #1 suggestions. For clarifications, see the figures below.

[Figure]

[Figure]

Thank you for detailed and thoughtful review comments that certainly was a good support for us to improve the ms.

---

## Author Comment (AC2)

**Author response to reviews and resulting revisions of ms egusphere-2025-4756: "Methane releases across the Laptev Sea signaled by time-integrated biomarkers of aerobic methane oxidation"**

**Reference: https://doi.org/10.5194/egusphere-2025-4756**

Albin Eriksson, Birgit Wild, Wei-Li Hong, Henry Holmstrand, Francisco Jardim de Almada Nascimento, Stefano Bonaglia, Denis Kosmach, Igor Semiletov, Natalia Shakhova, and Örjan Gustafsson.

We gratefully thank the referees for thoroughly reviewing and giving constructive feedback that is helping to clarify the significance and importance of this manuscript during revision.

All reviewer comments are included below in *black italic font* each followed by our detailed author responses, formatted as indented blue text. Citations of our implemented changes in the manuscript are formatted as indented italic blue text.

*Anonymous Referee #2:* [https://doi.org/10.5194/egusphere-2025-4756-RC2](https://doi.org/10.5194/egusphere-2025-4756-RC2)

*Elevated methane concentrations in seawater have been widely reported in the Eastern Siberian Arctic seas underlying subsea permafrost. These methane concentrations show strong spatial and temporal variability.*

*In this study, the authors combine hopanoid-specific carbon isotope measurements ($\delta13C$- C30) with 16S rRNA gene analyses of surface sediments to trace aerobic methane oxidation (AeMO) as a proxy for methane release in the Laptev Sea. Depleted $\delta13C$- C30 values, ranging from −57 to −37‰, are interpreted as diagnostic of AeMO. The results suggest that methane release is most intense in the outer shelf region (OLS), consistent with previously reported seawater methane concentrations. Notably, depleted $\delta13C$- C30 values were also observed in the mid-shelf region (MLS), an area traditionally considered to exhibit low methane emissions, alongside methane concentrations that have not been reported previously. In contrast, high methane concentrations were measured near the Lena River delta in the inner shelf region (ILS), yet hopanoids there display comparatively heavier $\delta13C$- C30 values, indicating that different processes may be influencing the isotopic signal. The authors acknowledge the additional complexity this introduces for interpretation but argue that $\delta13C$- C30 values remain lower than bulk organic carbon $\delta13C$, supporting the presence of AeMO.*

*Main findings:*

1. *The mid-shelf region (MLS) exhibits higher methane concentrations than previously reported for this area.*

2. *The outer shelf region (OLS) is characterised by elevated methane concentrations and the most depleted $\delta13C$- C30 values, consistent with intense aerobic methane oxidation.*

3. *The inner shelf region (ILS) shows high methane concentrations, but comparatively heavier $\delta13C$- C30 values than the other regions, suggesting that additional processes influence the isotopic signal in this area.*

*This study presents interesting results on methane cycling in the Laptev Sea across the outer, mid-, and inner-shelf regions with very interesting trends and variations. The manuscript is generally well written and structured and the results show interesting findings. I have a small number of minor to major comments which, once addressed, it can help strengthen the interpretation.*

> We thank referee #2 for the overall support and positive words on the manuscript quality. Additionally, we thank the referee for comments and suggestions that helped improve both the clarity and the interpretations of the manuscript. In particular, the suggestion to provide literature comparisons of $\delta^{13}C$- $C_{30}$ hopenes, although they are small in number in marine surface sediments, have been made and added to the revised ms.

1. ***Major comments***

- *Hopanoids are not unique to methanotrophic bacteria. Although this study includes measurements of bulk δ13C-OC and presents a stable carbon isotope mixing model with defined end members, the interpretation would be strengthened by more explicitly contextualising the δ13C- C30 values within the existing AeOM literature. In particular, it would be helpful to include reported δ13C- C30 ranges from other methane seep studies where hopanoids have been used as proxies for methane release. This information could be included in both the Introduction and then in the discussion, comparing results with existing literature values. In addition, providing δ13C- C30 ranges from related environments, such as peatlands and lacustrine systems which are mentioned in the Introduction, would offer useful broader context. By providing this broader context it will make it easier to assess the AeOM results and improve the clarity and interpretability of the results throughout the manuscript.*

  We appreciate the suggestion and recognize the possibility to compare with application of similar approaches to other systems for a wider comparison of the here reported $\delta^{13}$C-C$_{30}$ hopenes. However, the current literature of $\delta^{13}$C-C$_{30}$ hopenes is largely limited to palaeoclimatological investigations in marine/lake sediments and shale deposits (e.g., Hinrichs, 2001, *Geochemistry, Geophysics, Geosystems*; Hinrichs et al., 2003, *Science*; Birgel and Peckmann, 2008, *Org Geochem*; Sun et al., 2022, *Nat Commun*; Blumenberg et al., 2024, *Paleoceanogr Paleoclimatol*; Yan et al., 2025, *Sci. Adv.*). The only other "contemporary" signals of AeOM using $\delta^{13}$C-C$_{30}$ hopenes are from peatlands (Inglis et al., 2019, *Geochim. Cosmochim. Acta*) and lacustrine sediments (Davies et al., 2016, *Biogeosciences*). Hence, comparisons with other contemporary "seep" environments are limited to non-marine settings.

  Nevertheless, after the referee suggestion, we have now revised the ms and provided reference values from Davies et al. (2016, *Biogeosciences*) and Inglis et al. (2019, *Geochim. Cosmochim. Acta*) to provide a comparison with our measured $\delta^{13}$C-C$_{30}$ hopene values:

  "*Generally, a larger presence of CH$_4$ has been linked to lower $\delta^{13}$C values of hopanoids (e.g., Inglis et al., 2019; van Winden et al., 2020; Yan et al., 2025). Therefore, $\delta^{13}$C-hopanoids may be used as indicators of the intensity of a time-integrated CH$_4$ release signal. In particular, hopanoid chain lengths ≤C$_{30}$ are generally more depleted in $^{13}$C compared to C$_{31}$-hopanoids and are therefore closely associated with MOB production of hopanoids (Inglis et al., 2019). Contemporary system calibrations of ≤C$_{30}$ hopanoids indicating AeOM exist in peatlands (≤C$_{30}$ hopanoids between -21 to -45 ‰; Inglis et al., 2019) and lacustrine systems (diploptene ranging from -38.8 to -68.8‰; Davies et al., 2016).* "*

- *In relation, the introduction would benefit from more clearly outlining previous studies that have applied the δ13C- C30 proxy to infer methane release, including a brief summary of their main findings. At present, it is not entirely clear whether this proxy is well established or in early stages, only one study is mentioned (van Winden et al., 2020. This ambiguity comes from the third paragraph of the introduction, which is a key section for framing the proxy and*

*one of the most important paragraphs in the manuscript. It would benefit from being rewritten.*

> We agree that the introduction text on this was too brief and this we have revised the ms to elaborate in greater detail in the Introduction on how well developed this proxy is to infer methane release and what has been accomplished to date. After the referee's suggestion, we have now provided a clarification that the proxy has been used to infer enhanced methane cycling in past climates (e.g., Hinrichs, 2001, *Geochemistry, Geophysics, Geosystems*; Hinrichs et al., 2003, *Science*; Birgel and Peckmann, 2008, *Org Geochem*; Sun et al., 2022, *Nat Commun*; Blumenberg et al., 2024, *Paleoceanogr Paleoclimatol*; Yan et al., 2025, *Sci. Adv*) and to asses contemporary methane cycling (Davies et al., 2016, *Biogeosciences*; Inglis et al., 2019, *Geochim. Cosmochim. Acta*) – and what conclusions have been possible to draw based on this proxy. The revised ms now also highlights that our manuscript is the first large-scale use of the proxy in marine surface sediments. The draft revised text for the second part of the paragraph now highlights the lack of modern system calibrations of the proxy in marine sediments and is described below:

> *"Generally, a larger presence of $CH_4$ has been linked to lower $\delta^{13}C$ values of hopanoids (e.g., Inglis et al., 2019; van Winden et al., 2020; Yan et al., 2025). Therefore, $\delta^{13}C$-hopanoids may be used as indicators of the intensity of a time-integrated $CH_4$ release signal. In particular, hopanoid chain lengths $\leq C_{30}$ are generally more depleted in $^{13}C$ compared to $C_{31}$-hopanoids and are therefore closely associated with MOB production of hopanoids (Inglis et al., 2019). Contemporary system calibrations of $\leq C_{30}$ hopanoids indicating AeOM exist in peatlands ($\leq C_{30}$ hopanoids between -21 to -45 ‰; Inglis et al., 2019) and lacustrine systems (diploptene ranging from -38.8 to -68.8‰; Davies et al., 2016). However, the lack of a large-scale comparison of $CH_4$ and hopanoids in marine systems leaves uncertainties for interpreting $\delta^{13}C$-hopanoids in geological records. In summary, $\delta^{13}C$-hopanoids in oxygenated surface sediments can be an informative tool to constrain a time-integrated $CH_4$ release signal, complementing observations of the highly variable $CH_4$ concentrations in the water column. "*

- *Anaerobic methane-oxidising archaea (ANME), which are often key components of methane seep ecosystems, were not investigated in this study. While ANME do not produce hopanoids, they commonly dominate methane oxidation in anoxic sediments and are therefore critical for understanding methane cycling in methane seep environments. Their absence may have important implications for the interpretation of the results, particularly in the inner shelf region (ILS), where high methane concentrations coincide with comparatively heavier δ13C-C30 values. Without information on sediment redox conditions or the presence and activity of ANME, it is difficult to determine whether reduced AeOM signals in this area reflect lower methane release, a shift toward anaerobic methane oxidation, or differences in carbon source mixing. Expanding the discussion to acknowledge this limitation would strengthen the overall interpretation of methane seep dynamics. In particular to consider how ANME-related*

*processes (potentially constrained by biomarkers such as archaeol or crocetane, or by microbial community data) might influence the observed patterns,.*

> We agree with referee #2 on the wider and general importance of AOM and ANME in seep environments. However, we stress that our study is focused on tracing the methane that is released to the aerobic systems (and thus not to deeper sediments where AOM and ANME occurs); with the current scope we use oxygenated surface sediments with the hopanoids as proxies of methane released into the aerobic water column of the region. Therefore, we do not present any results on the AOM through biomarkers such as archaeol, PMI or crocetane as that is outside the ms scope. For further information on the oxygen concentrations in the Laptev Sea water column and sediments see Stepanova et al. (2017, *Oceanology*), Brüchert et al. (2018, *Biogeosciences*), Xie et al. (2023, *Front. Mar. Sci).*and Maciute et al. (2025, *Env. DNA*).

- *In line of the points raised above, the sections discussing the inner shelf region (ILS) would benefit from reconsideration and restructuring. Based on the δ13C- C30 values reported for ILS, the AeOM signal appears relatively weak. Values that are only moderately depleted (e.g., not more negative than ~–40‰) quite likely reflect mixing between multiple bacterial carbon sources rather than a distinct methanotrophic signature, making it difficult to draw firm conclusions. Strengthening this discussion would likely require clearer consideration of (i) the relative roles of aerobic versus anaerobic methane oxidation, (ii) potential mixing with terrestrially derived organic matter delivered by the Lena River, and (iii) the limitations of the δ13C- C30 proxy in this specific setting where you have a large river. In addition, the interpretation in terms of MOB I versus MOB II would be more convincing if it was explicitly contextualised using the δ13C- C30 literature ranges as well as typical values for non-methanotrophic bacteria. Providing these comparative ranges would make it much easier for readers to assess whether the ILS signal is consistent with methanotrophy or more likely reflects mixed sources.*

> We thank referee #2 for suggesting to restructure section 4.2.2 and to further clarify the possible explanations for the moderately depleted $\delta^{13}$C-hopanoids in the ILS compared to the other regions of the Laptev Sea. As this manuscript focuses on hopanoids as a tracer of aerobic methane oxidation in surface sediments and water column, there is less need to elaborate on anaerobic methane oxidation because both the water column and surface sediments in the Laptev Sea are well oxygenated (Stepanova et al., 2017, *Oceanology*, Brüchert et al., 2018, *Biogeosciences*, Xie et al., 2023, *Front. Mar. Sci).*and Maciute et al., 2025, *Env. DNA*).

> Regarding the mixing from terrestrial sources, this is already explored in the submitted ms. The ms already acknowledges the importance of multiple bacterial sources as a possible reason for more enriched $\delta^{13}$C-hopanoids through our implementation of a Bayesian stable isotope mixing model (section 2.5 and equations 2-4) and the clear distinction of different isotope endmembers in Fig.3. Moreover, in Fig. 4 the relative abundance of 16S-rRNA displays the importance of "Other hopanoid producers" versus MOB-I and II. Additionally, we want to stress that the "moderate depletion of $\delta^{13}$C-hopanoids" in the ILS also likely is be a result from lower isotope

fractionation when MOB-II produce hopanoids and thus not necessarily only a dilution from non-methanotrophic sources. To conclude section 4.2.2. we already acknowledge these points in the submitted ms in lines 403-407 by a combination of three possibilities, but not exclusively one or the other:

*"Taken all together, the relative enrichment of $\delta^{13}$C-hopenes in the ILS compared to the OLS hotspot is likely a combination of: (1) a higher relative input of hopanoids from MOB-II, (2) non-methanotrophic bacteria produced in situ and in terrestrial settings and (3) lower activity of AeOM compared to the OLS. Nonetheless, the depleted $\delta^{13}$C-hop-17(21)-ene does indicate the presence of AeOM and thereby CH$_4$ release (Fig.3). "*

While mixed bacterial sources complicates the interpretation of a strictly AeOM related hopanoid signal, it is important to highlight the presence of MOB strictly related to marine environments. Therefore, we have revised our discussion in section 4.2.2 to clarify this in the ms:

*"Methyloceanibacter constituted the only genera of MOB-II in the ILS (Fig. S1) and has to this date to our knowledge only been isolated from marine systems (Takeuchi et al., 2014, 2019; Vekeman et al., 2016). Thus, the MOB-II in our study is likely an in situ signal of AeOM rather than of coastal influence as was found in the Kara Sea/Yenisei River (de Jonge et al., 2016), despite MOB-II generally dominating terrestrial ecosystems (Hanson & Hanson, 1996; Inglis et al., 2019). The less depleted $\delta^{13}$C-C$_{30}$ hopenes, yet in similar concentrations to the OLS, can therefore be an indication of MOB-II and a proxy-derived CH$_4$ signal in coastal regions needs thorough system knowledge to depict the source. "*

1. *The results appear to show a consistent geographic trend from the inner shelf region (ILS) to the outer shelf region (OLS), with organic carbon concentrations decreasing, δ13C- OC values becoming less negative (from approximately −26‰ to −23‰), and δ13C- C30 values becoming more depleted (from ~−39‰ to ~−52‰). The manuscript would benefit from a more integrated discussion of these spatial trends and the processes that may control their origin, as this could help unify the Results and Discussion sections and strengthen the overall interpretation.*

We agree and thank referee #2 for acknowledging these off-shelf trends and the importance of higher terrestrial input of organic carbon in the ILS. We have revised a paragraph in the new ms to discuss his trend in section 4.2.2 to highlight e.g., the higher terrestrial carbon content of the bulk organic carbon:

*"This study thereby indicates that a dilution from non-methanotrophic terrestrial sources may contribute to the higher $\delta^{13}$C-C$_{30}$ hopenes, due to the vicinity of the Lena River delta and higher relative abundance of non-methanotrophic hopanoid producers compared to MOB. This is further strengthened by $\delta^{13}$C-OC around ~26‰, indicating a larger terrestrial loading to the sediments of the ILS. Hopanoid source apportionment displayed that 77±7 % of hopanoids in the ILS are related to non-methanotrophic hopanoid synthesis and 16S-rRNA data indicates that the non-methanotrophic*

*hopanoid producers partly are of terrestrial origin. Consequently, the presence of hopenes more enriched in $^{13}C$ is also an indication of an additional contribution from non-methanotrophic hopanoid sources (party of terrestrial origin), rather than reflecting lower $CH_4$ releases from this region. "*

2. **Line by line comments**

*Introduction*

*Line 71- 73: This sentence is quite long and could be rephrased for clarity, as it is currently difficult to follow the main point. More generally, this paragraph would benefit from including a reported range of δ13C- C30 values for AeOM from the literature, which would help clarify how AeOM is being diagnosed in this study.*

This sentence has been divided into two sentences as follows:

*"Thus, hopanoid analyses are frequently combined with compound-specific isotope analysis of stable carbon isotopes ($\delta^{13}C$) to relate these biomarkers to $CH_4$ cycling. The isotopically depleted, $\delta^{13}C$-$CH_4$ (as low as -90 ‰; Milkov & Etiope, 2018) is used to differentiate $CH_4$-derived hopanoids from other sources (e.g., Hinrichs, 2001; Hinrichs et al., 2003; Birgel and Peckmann, 2008; Davies et al., 2016; Inglis et al., 2019; Sun et al., 2022; Blumenberg et al., 2024; Yan et al., 2025)."*

Ranges of $\delta^{13}C$-$C_{30}$ hopenes reported in other contemporary records, yet from other type of systems, have been included as follows:

*"Contemporary system calibrations of ≤$C_{30}$ hopanoids indicating AeOM exist in peatlands (≤$C_{30}$ hopanoids between -21 to -45 ‰; Inglis et al., 2019) and lacustrine systems (diploptene ranging from -38.8 to -68.8‰; Davies et al., 2016)."*

Please note that to help the reader understand how the $\delta^{13}C$-$C_{30}$ hopenes are interpreted in marine systems, there is in the submitted ms both careful and detailed text provided (4.2.1- 4.2.3) and an explanation by the graphical inset in Fig 3.

*Line 75: This sentence illustrates the point above. When stating that "a larger presence of methane has been linked to decreasing $\delta^{13}C$ values of hopanoids," you need to specify the $\delta^{13}C$–$C_{30}$ range associated with high methane presence, based on published studies.*

We thank referee #2 for the suggestion. As espoused and detailed at several locations above, common $\delta^{13}C$-$C_{30}$ hopenes from lacustrine and peatland settings have been added as comparisons due to the lack of $\delta^{13}C$-$C_{30}$ hopenes analyses in modern marine surface sediments. Please see the comments above.

*Line 76: The text mentions that this approach is "generally used," but only one relatively recent reference (Van Winden et al., 2020) is cited. Including additional references would help support the idea that this is a well-established and widely applied method.*

Additional references have been added as follows:

*"Generally, enhanced CH₄ cycling has been linked to decreasing δ¹³C values of hopanoids (e.g., Inglis et al., 2019; van Winden et al., 2020; Yan et al., 2025)."*

*Line 76: The phrase "intensity of time-integrated" could benefit from a brief clarification. A short explanation or example would help readers better understand what is meant here.*

This sentence has been changed to improve the clarity of "*time-integrated*" as follows:

*"Therefore, δ¹³C-hopanoids may be used as time-integrated indicators of enhanced CH₄ cycling."*

*Line 79: Introduce with few words or a short sentence the concept of contemporary system calibrations for hopanoids and why are they important for interpreting geological records.*

The sentence has been clarified as follows:

*"However, the lack of a large-scale comparison of CH₄ and hopanoids in marine systems leaves uncertainties when interpreting contemporary CH₄ cycling from reported δ¹³C-hopanoids in geological records."*

*Line 96: Since time-integrated proxies appear to be a central concept in this study (and are also linked to the point raised in line 76), it would be very helpful to introduce and explain this concept more explicitly earlier in the manuscript.*

The time-integration over several years has now been changed and clarified as follows:

*"In summary, δ¹³C-hopanoids in oxygenated surface sediments can be an informative tool to constrain a time-integrated CH₄ release signal on a years-decade scale, complementing observations of the CH₄ concentrations in the water column which is highly variable over much shorter timescales."*

*Line 97: OLS and ILS are mentioned here, but their locations are not clear from the map. It would be helpful to indicate both OLS and ILS in Figure 1.*

All subregions of the Laptev Sea (ILS, OLS, MLS) have now been added to each figure in the manuscript.

*Line 99: When referring to "high concentrations," specify what is being measured (e.g. methane concentrations) for clarity.*

High concentrations of biomarkers have been specified according to the referee's suggestion, as follows:

*"We hypothesize that known CH₄ ebullition hotspots in the Outer Laptev Sea and Inner Laptev Sea (OLS and ILS), where >1000 nM dissolved CH₄ concentrations have been observed (Shakhova et al., 2010, 2014; Steinbach et al., 2021), have high concentrations of C₃₀ hopanoids (diploptene, hop-17(21)-ene, neohop-13(18)-ene and diplopterol) with low δ¹³C values, tracing AeOM."*

*Line 99: The term "low δ¹³C values" is used here and throughout the manuscript, but it remains somewhat ambiguous. It may be clearer to indicate how low is low, or to refer directly to typical*

*δ13C- C30 values for AeOM. Including a short description of the $\delta^{13}C–C_{30}$ range for AeOM earlier in the introduction, based on literature values, would be very helpful.*

> Yes, we agree and as detailed at multiple places above, the common $\delta^{13}C\text{-}C_{30}$ hopenes associated with AeOM from lacustrine and peatland settings will be elaborated to a greater extent as comparisons due to the lack of $\delta^{13}C\text{-}C_{30}$ hopenes analyses in modern marine surface sediments.

*Line 100: It is not immediately clear where the mid-shelf is located relative to OLS and ILS. Referring explicitly to Figure 1 would help orient the reader.*

> The location of each region (ILS, MLS, OLS) has now been more clearly indicated in each figure.

*Line 101: The second hypothesis is somewhat unclear. When referring to "lower concentrations of higher hopanoids," it is not clear whether this refers to lower $\delta^{13}C–C_{31}$ values. Since the manuscript focuses primarily on $\delta^{13}C–C_{30}$, this section may benefit from clarification and consistency in terminology.*

> The terminology of this section has been clarified, as follows:

> "*In contrast, we hypothesize that the mid-shelf region without any discovered $CH_4$ hotspots, yet with dissolved methane in the range 10-60 nM (Fig. 2) display lower concentrations of $C_{30}$ hopanoids more enriched in $^{13}C$*".

*Methods*

*Line 113: As mentioned earlier, OLS and ILS should be clearly indicated on the map.*

> The location of each region (ILS, MLS, OLS) has now been indicated clearly in each figure.

*Line 158: While the sampled interval (1–2 cm slice) is provided, it would be helpful to also report the mass of material used for δ13C- OC analyses.*

> The mass of sediment used for $\delta^{13}C\text{-}OC$ analyses has been added.

*Line 160: Please clarify what is meant by "Ag capsules." If "Ag" is an abbreviation, it should be defined at first use.*

> Ag capsules has been changed to "*silver capsules*".

*Line 218: Indicate how much sample material was used for 16s rRNA analyses*

> The typical mass used for 16S-rRNA analyses has been added.

*Line 235: The absence of analyses targeting anaerobic methane-oxidising archaea (ANME) may need further discussion. While ANME does not produce hopanoids, they are commonly present in anoxic methane seep environments and can be useful for identifying oxic–anoxic transitions in sediments. As oxygen data is not available for the sediment cores, it is possible that areas with lower AeOM signals reflect more anoxic conditions dominated by ANME rather than lower methane release. Considering ANME (and associated biomarkers such as archaeol and crocetane) alongside hopanoid-based proxies would strengthen the interpretation of methane seepage intensity.*

We have in earlier comments acknowledged the general importance of AOM in seep environments and explained why this system, deeper into the sediments, is not the study system of the current investigation. We highlight that we through the hopanoids trace AeOM in the oxygenated surface sediments and aerobic water column. Thereby, we are focusing on the methane released to the water column and therefore we do not investigate biomarkers that trace AOM at greater sediment depths.

*Line 245 to 265: For the statistical analyses, it would be helpful to clearly outline throughout the manuscript (results, discussion, conclusions) that these estimates are semi-quantitative, not only here.*

We revised and highlighted that the isotope mixing model weighted against 16S-rRNA is semi-quantitative.

*Results*

*Line 283: The results suggest that OC concentrations decrease from ILS to MLS to OLS. If this reflects the geographic order, presenting the results consistently in that sequence will improve clarity of the manuscript.*

The order of presenting the results has been changed throughout to 1) OLS, 2) MLS and 3) ILS to stay consistent with the rest of the results.

*Line 285: Same for δ13C.*

See above. The order of presenting the results has been changed to 1) OLS, 2) MLS and 3) ILS to stay consistent with the rest of the results.

*Line 290 to 294: There is a relatively large uncertainty associated with OLS values, can you provide an explanation? Perhaps in the methodology. Additionally, the statement that OLS and ILS have similar concentrations, while MLS and ILS show no significant difference, is somewhat confusing and could be rephrased for clarity.*

We thank referee #2 for suggesting a clarification of the presented variability. The large range of hopanoid concentrations in the OLS is likely related to the presence of main "seep-stations" with pronounced ebullition and extremely high methane concentrations (and related hopanoid production) causing a log-normal distribution of the data, which explains the large standard deviation of hopanoid concentrations in the OLS.

The sentence describing the hopanoid concentrations in the OLS and the ILS have been revised for clarity according to the referee suggestions as follows:

*"The highest concentrations were found in the OLS ($35\pm20$ µg $gOC^{-1}$; n=15; Fig.5). Relatively high $\sum C_{30}$-hopenes concentrations were also present in the ILS, and showed no significant difference compared to the OLS ($18\pm11$ µg $gOC^{-1}$; n=6; Fig.5; Supplementary Table 4)."*

The sentence describing the hopanoid concentrations in the MLS versus the ILS have been revised for clarity according to the referee suggestions as follows:

> *"In contrast, the concentrations of ∑C30-hopenes were significantly lower in the MLS (10±4 µg gOC-1 ; n=4; Fig.5) compared to the OLS, but showed no significantly different concentrations compared to the ILS (Supplementary Table 4)"*

*Line 303: If −57‰ represents the most depleted δ13C value observed at OLS stations, state more explicitly. You might also consider only reporting mean values with standard deviations, or instead providing ranges for OLS, MLS, and ILS, rather than listing individual station values. Whichever approach is chosen, stay consistent across the results section.*

> The presentation of the results has been revised according to the referee suggestion and now only includes averages and standard deviations, as follows.

> *"The $\delta^{13}C$-$C_{30}$-hopenes were lowest in the OLS with a mean±standard deviation of -52.9±4.3 ‰ across the region (Fig. 5)."*

*Discussion*

*4.2.2. ILS paragraph: This section would benefit from substantial revision. When discussing more or less depleted values, it is necessary to include specific δ13C- C30 values for ILS results. It would also be helpful to compare them with published literature values for AeOM signal diagnosis. The ILS appears to be the area where the δ13C- C30 proxy for methane release is least straightforward. Although methane concentrations are high, δ13C- C30 values do not seem to clearly indicate AeOM. Given the proximity of the ILS to the Lena River—one of the largest rivers globally—terrestrial carbon inputs may influence the δ13C- C30 signal. The link between δ13C- C30 values and methane release in this area should therefore be treated with caution.*

> We agree that the ILS is the most complex system and acknowledge the need for comparisons with published literature of $\delta^{13}C$-$C_{30}$-hopenes, as espoused in several author replies to this referee comment already above. As mentioned above, to our knowledge the contemporary climate calibrations of these biomarkers are limited to peatland (Inglis et al., 2019, *Geochim Cosmochim Acta*) and lacustrine settings (Davies et al., 2016, *Biogeosciences*). Nevertheless, we have included for comparison the range of $\delta^{13}C$-$C_{30}$-hopenes observed in Davies et al. (2016, *Biogeosciences*) and Inglis et al. (2019, *Geochim Cosmochim Acta*) in section 4.2.2.

> Regarding the role of terrestrial carbon input, we have elaborated on in this in a comment above and incorporated a clear distinction in the revised ms. Additionally, the terrestrial loading from the Lena River was also already included in the submitted ms and in detail discussed in section 4.2.2, and lines 121-122, and 394-395. However, as we also replied to referee #1, we want to again highlight that it is clear that the methane source is from the coastal system and not the river as very high methane concentration gradients, intensive bubbling, and much lower concentrations in the river waters have been observed (e.g., Shakhova et al., 2007, *J. Mar. Sys*; Shakhova et al., 2010, *Science; Shakhova et al.,* 2014, *Nat. Geo.*; Shakhova et al., 2017, *Nat. Comm*).

*Line 379: How can it be ruled out that lower AeOM activity in the ILS is not driven by more anoxic sediment conditions and a higher contribution from ANME, rather than reduced methane release? If there is data available from the literature for sediment oxygen levels or ANME presence/absence, this section would benefit of a small discussion here in order to rule this out (or not).*

We want to highlight that a less depleted $\delta^{13}$C-$C_{30}$proxy-signal does not suggest lower methane release from this region (as evidenced by the high methane concentrations). The surface sediments in this region are not anoxic, as we pointed out in detail a comment above. No indications of known ANME or nitrate dependent methane oxidizers such as *Methylomirabilis oxyfera* producing hopanoids was found in the 16S-rRNA data. Additionally, we stress that the $\delta^{13}$C-$C_{30}$-hopenes trace AeOM in oxygenated surface sediments and the water column, so the proxy indicates methane release into the oxygenated water column and not the processes within the sediment anoxic sediments.

The submitted ms already includes statements displaying that these surface sediments and the water column are well oxygenated in lines 85-86, 110-111 and 345-346.

*Line 390: Including literature values for typical δ13C- C30 values of hopanoids produced by non-methanotrophic bacteria would help clarify how mixing between sources may influence the observed signal.*

Typical literature values for $\delta^{13}$C- $C_{30}$ values in modern lacustrine and peatland settings have been included (see comments at multiple places above).

*Conclusion*

*The conclusions might benefit from more cautious wording, emphasising that the δ13C- C30 – methane release proxy appears robust for MLS and OLS, but that in areas influenced by large terrestrial inputs (such as the Lena River), this proxy likely needs to be complemented by additional lines of evidence.*

We agree. Considerations when interpreting $\delta^{13}$C- $C_{30}$ values in coastal settings have now been included in the revised ms.

1. ***Minor comments:***

- *Several sentences throughout the manuscript are quite long (three lines or more). Breaking these into shorter sentences would improve readability.*

Sentences throughout and especially in the Conclusion section have been shortened.

- *I suggest being coherent when describing ILS/MLS/OLS. For clarity it would be helpful if you follow the geographical progression*

The order of discussion has been changed to ensure consistency throughout the manuscript.

- *Line 301: "Strikingly" may sound somewhat strong; "remarkably" could be a suitable alternative.*

Strikingly has been changed to "remarkably".

- *The discussion section header titles are confusing, re-name to clearly indicate you are discussing: 1. OLS, 2. ILS, and 3. IMS. Similarly to above, I would suggest to do in geographical order.*

We will carefully revisit the section header titles and ensure the reflect the content of their respective sections.

- *It would be helpful to clearly define key terminology early in the Introduction (e.g. $\delta^{13}C$ –OC, $\delta^{13}C$ –$C_{30}$, $\delta^{13}C$–$CH_4$).*

While $\delta^{13}C$ and CH4 are common abbreviations/notations in Biogeoscience, we have revised the ms to introduce also these type of terminology at first usage.

- *All maps should include scales, and OLS/MLS/ILS areas should be clearly indicated.*

  All maps include lat/lon, see detailed comment above.

1. **Figures:**

*Figure 1: Indicate ILS, MLS, and OLS areas in the map. The coordinate labels are very large, they can be reduced in size, and then the rectangular map enlarged. Label the Lena River and add a scale for the map. The land–sea boundary is currently difficult to interpret, maybe changing the map/permafrost overlay could help.*

> All subregions of the Laptev Sea (ILS, OLS, MLS) have now been included in each figure. Regarding adding a scale bar to each figure, we are more skeptical. First, it would make busy figures even more crowded. Secondly, a scale of e.g. 1 cm on the figure represent different distances at different latitudes so this would not be accurate. We believe readers are familiar with the concepts of Lat and Lon and that this scale will suffice.

*Figure 2: A scale bar should be included, reduce panel labels (a, b, c, d) size. The methane concentrations colour scale makes it difficult to distinguish values between ~50 and 300 nM; adjusting the colour scheme may help. Indicating OLS, MLS, and ILS on the map and summarising key spatial trends in the caption would improve interpretability.*

> The panel sizes have been included and the color gradients have been changed to ensure better visibility. A new scale bar for concentrations has been added. Please see drafts of revised figures below.

[Figure]

*Figure 3: Panel labels (a, b, c) should be included directly in the figure, not only in the caption. The shaded area referred to as "grey" in the caption appears green in the figure and should be made consistent. Please also indicate the literature sources of the hopanoid end-member values in the caption and clarify that the shaded areas are based on semi-quantitative estimations.*

Panel labels have now been included in the figure. All the literature sources of the endmembers are now included both in supplementary tables S6 and S7 and in the figure 3 caption.

Thank you for detailed and thoughtful review comments that certainly is a good support for us to improve the ms.